# Elastocapillary cleaning of twisted bilayer graphene interfaces

Yuan Hou[1,2,6], Zhaohe Dai[3,6], Shuai Zhang [ID] [4,5,6], Shizhe Feng[4], Guorui Wang[1,2], Luqi Liu [ID] [1✉], Zhiping Xu[4✉], Qunyang Li [ID] [4,5✉] & Zhong Zhang [ID] [1,2✉]

Although layered van der Waals (vdW) materials involve vast interface areas that are often subject to contamination, vdW interactions between layers may squeeze interfacial contaminants into nanopockets. More intriguingly, those nanopockets could spontaneously coalesce into larger ones, which are easier to be squeezed out the atomic channels. Such unusual phenomena have been thought of as an Ostwald ripening process that is driven by the capillarity of the confined liquid. The underlying mechanism, however, is unclear as the crucial role played by the sheet's elasticity has not been previously appreciated. Here, we demonstrate the coalescence of separated nanopockets and propose a cleaning mechanism in which both elastic and capillary forces are at play. We elucidate this mechanism in terms of control of the nanopocket morphology and the coalescence of nanopockets via a mechanical stretch. Besides, we demonstrate that bilayer graphene interfaces excel in self-renewal phenomena.

[1] CAS Key Laboratory of Nanosystem and Hierarchical Fabrication, CAS Center for Excellence in Nanoscience, National Center for Nanoscience and Technology, Beijing, P.R. China. [2] CAS Key Laboratory of Mechanical Behavior and Design of Materials, Department of Modern Mechanics, University of Science and Technology of China, Hefei, P. R. China. [3] Department of Aerospace Engineering and Engineering Mechanics, The University of Texas at Austin, Austin, TX, USA. [4] Applied Mechanics Laboratory, Department of Engineering Mechanics, and Center for Nano and Micro Mechanics, Tsinghua University, Beijing, P. R. China. [5] State Key Laboratory of Tribology, Tsinghua University, Beijing, P. R. China. [6] These authors contributed equally: Yuan Hou, Zhaohe Dai, Shuai Zhang. ✉email: liulq@nanoctr.cn; xuzp@tsinghua.edu.cn; qunyang@tsinghua.edu.cn; zhong.zhang@nanoctr.cn

Graphene and the plethora of other two-dimensional (2D) materials have shown exciting properties to new physics and a wide range of applications[1–3]. Recent studies indicate that these properties may be consolidated by the layer-by-layer assembly of the sheets into so-called van der Waals (vdW) materials[2–4]. For example, the twisted bilayer graphene (tBLG) has exhibited a host of new physical properties such as moiré bands, strongly correlated electrons, and superconductivity[5–7]. The vdW material not only exploits the functionalities of each sheet but also protects its functional performance from degradation (often caused by external impurities)[8,9]. Besides, the ultrahigh surface-to-volume ratio makes 2D materials exposed to adventitiously ambient contaminations (such as water and hydrocarbons) during the transfer process[10]. The associated interfacial vdW interactions could be substantial enough to drive these small molecules together into pockets (also known as bubbles)[10–12]. This phenomenon has been dubbed "self-cleaning" as it leaves the rest of the interface atomically clean[10].

Similar scenarios of thin solids confining liquids have been extensively investigated at larger scales, including liquid blisters and lenses[13,14]. What makes the vdW material system particularly intriguing is the mobility of the confined liquids. For instance, experiments have revealed that pockets could aggregate and merge into larger ones over time or after annealing in a spontaneous manner[15,16]. This coarsening behavior is practically useful because larger pockets of contaminations are more likely to be squeezed out of the atomic channel, for example, by a stamp[17,18]. A natural question that is more intriguing from a fundamental perspective is what drives the spontaneous aggregation and coalescence of those pockets. Previous studies have suggested an Ostwald ripening mechanism, which is a process of capillarity-driven mass diffusion from smaller to larger bubbles[19–21]. However, this mechanism neglects important aspects of thin sheet elasticity and cannot explain the ripening of remote pockets without substance transport (we shall see shortly).

Motivated by this question, we focus on one of the most promising vdW materials—tBLG. We use a suspended micro-nano bubble device to investigate the morphology of nanopockets formed within tBLG as well as study how they merge. We reveal the self-renewal of tBLG interface: (i) Nanopockets of water or ethanol contaminations can move readily and leave indiscernible residues; (ii) Interfacial topologies such as moiré patterns that are initially contaminated can recover. Furthermore, combining the elasticity of graphene sheet with the capillarity of trapped liquid, we proposed the elastocapillary cleaning mechanism.

## Results

**Moiré patterns and self-cleaning**. The micro-nano bubble experimental setup is illustrated in Fig. 1a. Monolayer graphene sheets are prepared by mechanical exfoliation of graphite. A water droplet-assistant transfer method is used to stack two separate monolayer sheets on a microhole-patterned $SiO_2$/Si substrate[22,23] (see details in the "Methods" section and Supplementary Fig. 1). By introducing the twist angle, two monolayer graphene sheets with a mismatch of the orientation lead to a hexagonal moiré pattern[24,25]. The vdW forces between sheets can trap water molecules at the graphene–graphene interface and spontaneously lump them together into water pockets. The pockets typically range tens to hundreds of nanometers in radius, as shown in Supplementary Fig. 1h. We note that similar configurations have been widely utilized to visualize liquid-phase matters in the field of liquid cell electron microscopy (EM)[26,27]. Unlike EM that is only able to characterize the projected in-plane dimensions, the experimental setup in Fig. 1a allows for the measurement of the three-dimensional geometry of nanopockets via atomic force microscopy (AFM).

We show AFM lateral force images of the suspended tBLG containing nanopockets in Fig. 1b–f. A striking feature comes from the periodic moiré patterns. The period $L$ can indicate the twist angle $\theta$ via $\sin(\theta/2) = a/(2L)$, where $a$ is the lattice constant of graphene (i.e., 0.246 nm)[28]. Accordingly, the twist angle of the sample in Fig. 1b–d is ~1.5° (corresponding to $L \approx 9.4$ nm), whereas ~0.6° ($L \approx 23.5$ nm) in Fig. 1e, f. Although previous works mostly reported relatively uniform moiré patterns[29–31], our experiments show that moiré patterns become distorted near water nanopockets and disappear inside. We use this behavior to determine the location of the nanopocket edge (i.e., the radius). The out-of-plane profile of the nanopocket is extracted from the simultaneously measured AFM height signals (see details in Supplementary Fig. 2).

The presence of moiré patterns outside the nanopocket regions implies the atomic cleanness of the graphene interlayer interface. We find that the nanopockets are very mobile: they can be actuated by tip scanning (see details in the "Methods" section). For example, the motion of a single nanopocket in Fig. 1b–d is caused by the contact-mode AFM scanning. The moving direction of the nanopocket is consistent with that of the tip, which sweeps horizontally while moving from the bottom to the top. Notably, the initially water-contaminated region ends up with perfect moiré patterns, suggesting that vdW interfaces between graphene sheets recover themselves to a contamination-free state.

We further focus on a region containing two neighboring nanopockets (Fig. 1e) and carry out AFM scans following the same experimental setting. Compared with the single nanopocket in Fig. 1b–d, the bottom nanopocket in Fig. 1e moves even faster as the tip entirely misses the nanopocket's trajectory. However, from Fig. 1f, we may infer that the bottom nanopocket travels toward the top one and merges into it spontaneously. The coalescence forms a larger nanopocket, again along with the recovery of the interface (labeled by the dashed circle in Fig. 1f). Through measurement of the morphology of 18 samples (Supplementary Figs. 3 and 4), we find that the ratio of the total volume of nanopockets before and after the coalescence mainly ranges from ~0.90 to ~1.05. Considering that the nanopocket after coalescence features reduced internal pressure[7,8], we suggest that nanopockets may mostly contain liquid water (Supplementary Fig. 5), and the amount of compressible substance (e.g., air) may be small.

It has been reported in the literature that nanopockets trapped at 2D material interfaces could coalesce over time or upon annealing, which has been thought of as an Ostwald ripening process[15,16,20]. Our finding suggests another useful aspect of coalescence: it can occur rapidly at room temperature in response to mechanical stimuli. More importantly, the experimental results in Fig. 1 reveal subtleties associated with the elastic deformation of the graphene sheet, including the bulging and wrinkling, respectively, inside and outside nanopockets. These observations highlight the crucial role of graphene elasticity. Such a role seemingly suggests an elastic version of the ripening mechanism beyond the traditional Ostwald ripening process between nanopockets (mainly considering mass diffusion)[15,16,20]. In particular, the nanopockets may interact remotely, as hinted by the interweaved wrinkling pattern in the region between the two nanopockets in Fig. 1e.

**Geometry of nanopockets in tBLG with various twist angles**. To understand the ripening mechanism, we begin by understanding the geometry of nanopockets. We focus on nearly round nanopockets whose ratio of the radii along the semi-minor to semi-

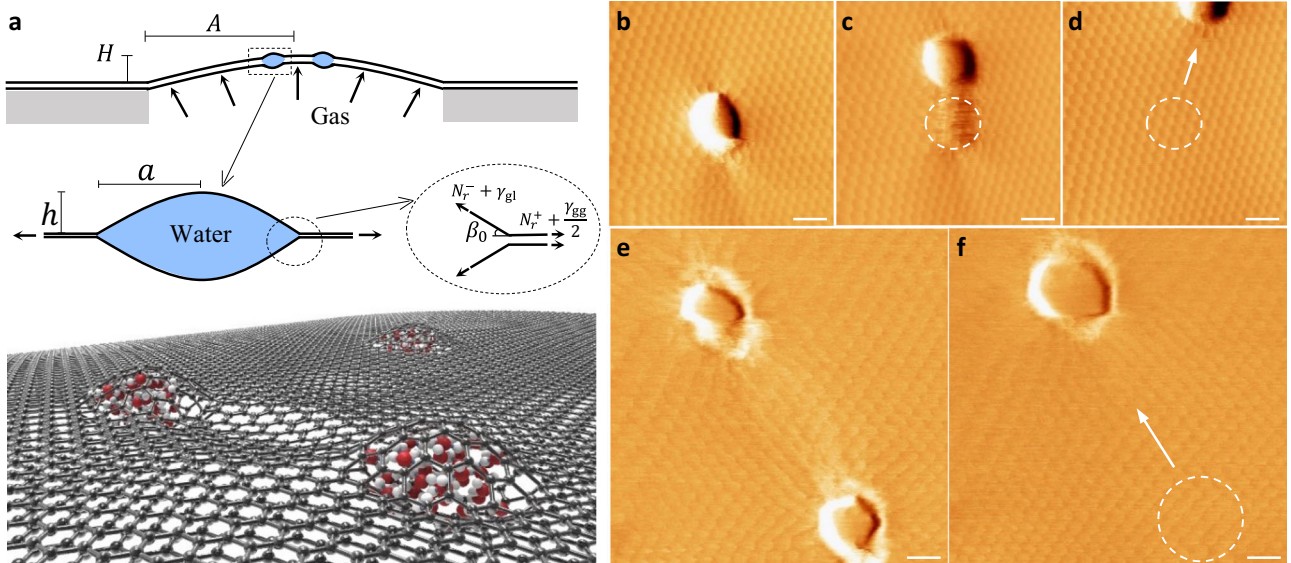

**Fig. 1 Device setup and demonstration of the self-cleaning phenomena. a** Schematic of twisted bilayer graphene (tBLG) suspended on a hole of radius $A$ (1.5 μm in our experiments). The tBLG can be pressurized by gas, which leads to a shallow spherical shell of height $H$. The zoom-in view shows a nanopocket of water of radius $a$ and height $h$ trapped at the bilayer interface (exaggerated). The edge of the nanopocket is enlarged twice to illustrate the force balance at the contact line. $\beta_0$ is the effective contact angle. $N_r^+$ and $N_r^-$ represent the radial sheet tension at the outer and inner edge. $\gamma_{gg}$ and $\gamma_{gl}$ are the energy density (per unit area) for the twisted graphene–graphene interface and the graphene–liquid interface, respectively. **b–d** Motion of a single nanopocket induced by the AFM probe. **e, f** Motion of a nanopocket when it is next to another one. These AFM lateral force images come from relatively flat tBLG (−10 nm ≲ H ≲ 10 nm). The scanning was performed repeatedly under contact mode, where the AFM tip sweeps on the graphene surface horizontally while moving from the bottom to the top. The dashed circles and the arrows indicate the initial locations and moving directions of nanopockets, respectively. Scale bars: 30 nm.

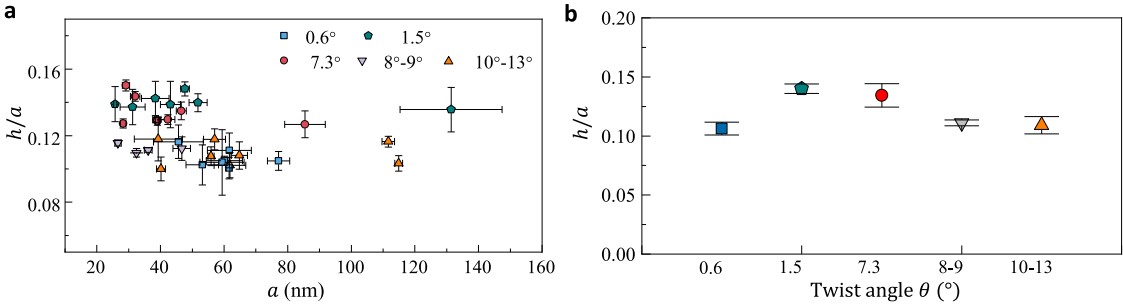

**Fig. 2 Aspect ratios of nanopockets in response to twisting. a** Radius-dependency of aspect ratios for nanopockets confined in tBLG with various twist angles. The twist angle was determined according to the period of moiré patterns for small-angle samples ($\theta \lesssim 7.3°$, Supplementary Fig. 7) and estimated based on the Raman spectra for large-angle samples (Supplementary Fig. 8). **b** Twist angle dependency of aspect ratios. The data were collected from **a**. For each nanopocket, the radius $a$ and bubble height $h$ were taken along three directions, from which the average values of $a$ and $h/a$ as well as x(y)-error bars were determined.

major ellipse axes is larger than 0.9 (Supplementary Fig. 6). It is found that a spherical cap could approximately characterize the out-of-plane profile of the nanopocket[12,32]. The primary geometric descriptors are the center height $h$ and the radius $a$, as labeled in Fig. 1a. We find that the aspect ratio ($h/a$) of nanopockets is not sensitive to the radius for a given twist angle of the host tBLG (Fig. 2a). This implies a geometric self-similarity or an invariant contact angle of the confined water, like the classical wetting behavior of water droplets on a solid surface[33]. Similar to the classical wetting problem, it is natural to consider the horizontal force balance at the contact line (the vertical balance is enforced already by the up–down symmetry) (Fig. 1a inset),

$$\cos\beta_0 = \frac{N_r^+ + \gamma_{gg}/2}{N_r^- + \gamma_{gl}}, \qquad (1)$$

where $\beta_0$ is the effective contact angle, $N_r$ is the radial sheet tension at the edge, $\gamma_{gg}$ and $\gamma_{gl}$ are the energy density (per unit area) for the twisted graphene–graphene interface and the graphene–liquid interface, respectively. Detailed analysis is presented in Supplementary Section 5. We assume that the jump of the slope of the sheet does not cause the discontinuity of the radial stress, i.e., $N_r^+ = N_r^-$ [33,34]. The simple form of Eq. (1) benefits from the almost vanished friction at the twisted graphene–graphene interface, which is termed superlubricity[35,36]. The essence of the nanopocket geometry is simply the contact angle being modified by the membrane tension.

To understand what determines such contact angle, we begin with elementary geometry. Both $\left(1 - \cos\beta_0\right)$ and the strain components scale as $h^2/a^2$ [37,38]. The radial stress then approximates as $E_{2D}h^2/a^2 + T_{pre}$ in the graphene sheet of elastic

stiffness $E_{2D}$ subject to a pretension $T_{pre}$. Experiments in Fig. 2b are based on relatively flat tBLG sheets where $T_{pre}$ could be negligible. It can be found that the aspect ratios of nanopockets are insensitive to the radius of nanopockets and twist angles of tBLG. The observed size-insensitive aspect ratio of nanopockets becomes clear if Eq. (1) is rewritten as

$$\left(\frac{h}{a}\right)^4 \sim \Delta\widetilde{\gamma} - \widetilde{\gamma}_{gl}\left(\frac{h}{a}\right)^2, \tag{2}$$

where $\widetilde{\gamma}_{gl} = \gamma_{gl}/E_{2D}$, and $\Delta\widetilde{\gamma} = (2\gamma_{gl} - \gamma_{gg})/E_{2D}$. The aspect ratio is usually small; it is reasonable to neglect the second term on the right side if $\widetilde{\gamma}_{gl}$ is not too greater than $\Delta\widetilde{\gamma}$. $\Delta\widetilde{\gamma}$ stands for the change of interfacial energies for the formation of a nanopocket with respect to the sheet stiffness and thus can be understood as an elastocapillary parameter[11,12], which should be $\sim\mathcal{O}(10^{-4})$. Systems with larger $\Delta\widetilde{\gamma}$ produce more significant elastic deformation or nanopockets with higher aspect ratios.

A unique feature of nanopockets in vdW materials is the existence of stress/strain fields outside the nanopocket. In particular, a tensile $N_r^+$ may come with compressive hoop stress $N_\theta^+$, which can further lead to the radial wrinkling of the sheet to relieve such compression. This explains the distorted moiré patterns surrounding nanopockets in Fig. 1. We will later attribute the pocket–pocket interaction to the extended stress fields outside nanopockets (behaving like long-range forces acting on one another). Before that, we discuss the rich physics in the pretension effect on the geometry of a nanopocket.

**Nanopocket geometry upon a stretch.** Since we suspended the host tBLG on round holes of radius $A$ (Fig. 1a), the pretension to local nanopockets can be introduced by pressurizing the tBLG drumhead with a height of $H$ via the well-established method[39,40] (see details in the "Methods" section and Supplementary Fig. 9). We focus on nanopockets located near the center of the bulged drumhead where the global pretension to these nanopockets is nearly equibiaxial of magnitude $T_{pre} \sim E_{2D}H^2/A^2$ [41]. Detailed numerical calculations are presented in Supplementary Material Section 5. We find that when the wrinkling is absent, the aspect ratio of a nanopocket subject to a pretension follows a simple form:

$$\left(\frac{h}{a}\right)^4 + \left(\frac{h}{a}\right)^2\left(\frac{H}{A}\right)^2 \cong \left(\frac{h}{a}\right)_0^4, \tag{3}$$

where the pretension-free aspect ratio $(h/a)_0 = \Delta\widetilde{\gamma}^{1/4}$ according to Eq. (2).

In Fig. 3a, we show aspect ratios of nanopockets sandwiched by tBLG with two sets of twist angles ($\theta \approx 7.3°$ and $10° \sim 13°$). The pretension is controlled by deforming the host bilayer drumhead with different $H/A$ (Fig. 1a). We find decent agreement between the prediction by Eq. (3) (solid curves) and measurements (markers) in Fig. 3a, where $\Delta\widetilde{\gamma}^{1/4}$ in the theoretical analysis is taken as ~0.122 and ~0.107 for tBLG with $\theta \approx 7.3°$ and $10° \sim 13°$, respectively. Apparently, the application of pretension flattens the nanopocket profile. Similar behavior has been recently observed in droplets capped with thin polymeric films, in which pretension overwhelms the tension associated with the out-of-plane deformation[14]. What is new in our experiments is that the latter ($\sim E_{2D}h^2/a^2$) is more likely to be greater than or at least comparable to the pretension ($\sim E_{2D}H^2/A^2$).

Eq. (3) becomes invalid when the pretension is not substantial enough to suppress instabilities. Numerical calculations provide a

criterion for the instability formation, namely

$$\left(\frac{h}{a}\right)^2 \geq \left(\frac{h}{a}\right)_{cr}^2 \cong 1.8\left(\frac{H}{A}\right)^2, \tag{4}$$

which dictates the shaded region in Fig. 3a. In experiments, the instabilities can be better seen in the phase panel of the AFM scanning, including Fig. 1b–f, Fig. 3b, and Supplementary Fig. 10a. We use crossed markers to denote nanopockets whose wrinkles have been suppressed by the applied pretension. Though our experimental data are relatively sparse regarding $H/A$, we find good agreement between the theoretically predicted criterion, i.e., Eq. (4) and dashed line in Fig. 3a and experiments. However, this criterion may slightly overstate the instability domain owing to the pre-existing pretension in the transfer process. For instance, some of the data are supposed to be in the shaded region in Fig. 3a, while the wrinkles have been suppressed.

To elucidate the aspect ratio of wrinkled nanopockets, we use tension field theory that completely relieves the hoop compression in the wrinkled region of the sheet (see details in Supplementary Material Section 6). This treatment should be valid for graphene owing to its thinness[42]. Though the calculations (dashed curves in Fig. 3a) successfully predict the increase of $h/a$ with the decrease of $H/A$ in the shaded region, they seem to overestimate the nanopocket aspect ratio. This implies that the associated elastic energies have been underestimated in our model. The deviation may be ameliorated by a more detailed analysis considering the initial pretension, the non-uniformity of wrinkling patterns, the effect of the drumhead curvature[14,43] as well as the "mechanics" arising when pretension is relatively weak[44].

Considering that in devices the vdW interfaces may be subject to contamination of various substances, we further explore a bit ethanol nanopocket (see preparation details in Methods Section). The comparison between water and ethanol nanopockets in morphologies and aspect ratios is shown in Supplementary Fig. 11. In brief, the wrinkled regions usually exist around the nanopockets when there is no pretension on the host tBLG, which is independent of the type of contaminants. Besides, the pretension response of the two types of nanopockets is found to be identical: as the pretension increases, the wrinkled region shrinks and finally disappears, which is shown in Fig. 3d, e. It is worth noting that the geometrical criterion in Eq. (4) still works well since it remains invariant for any specific contaminant (Supplementary Fig. 12). We suggest that the behaviors of these water and ethanol nanopockets are quite similar, which should be controlled by the same elastocapillary mechanism we presented.

**Merging of wrinkled and unwrinkled nanopockets.** Since the pretension could suppress the wrinkling, we are also interested in how the wrinkling pattern or pretension affects the merging process of nanopockets. Figure 4a–c shows the coalescence process of two neighboring ethanol nanopockets on the tBLG without pretension (so wrinkles exist). A mechanical stimulus was applied to the nanopocket-B from bottom to top. Similar to the results in Fig. 1, we find the nanopocket-B moves suddenly towards the nanopocket-A (Fig. 4b) and merges together after the second mechanical stimulus (in Fig. 4c). This observation seems to reveal an attractive force between the neighboring wrinkled nanopockets. The further coalescence process is shown in Supplementary Fig. 13. We use the same experimental methods to observe the coalescence of unwrinkled nanopockets under ~1% pretension (Fig. 4e–g). However, the actuated nanopocket-D moves along the direction of mechanical stimuli rather than merge with the nanopocket-C directly. This implies that the pocket–pocket attraction, like the wrinkling pattern, has been somehow suppressed by the pretension. From the

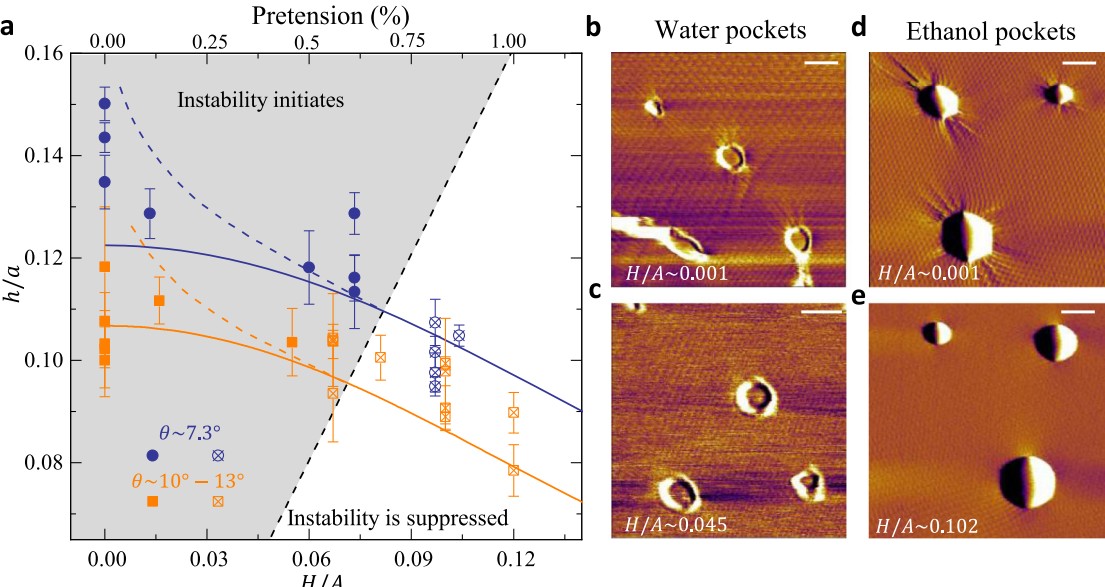

**Fig. 3 Aspect ratios of nanopockets in response to stretching. a** The dependence of the aspect ratio of water nanopockets on the aspect ratio of their host drumheads. Nanopockets whose aspect ratios are in the shaded region are predicted to be subject to hoop compression and hence elastic instabilities near the edge. Solid and dashed curves are from theoretical calculations without and with considering elastic instabilities of graphene sheets, respectively. Solid markers are obtained via pressurizing the drumhead. The cross in markers indicates the wrinkles around nanopockets are suppressed. The aspect ratio is measured based on nanopockets located in the center of tBLG drumhead so that the stretch to the nanopockets caused by the pressurization is relatively uniform. For each nanopocket, the radius $a$ and bubble height $h$ were taken along three directions, from which the average value of $h/a$ and y-error bars were determined. **b, c** AFM lateral force images of water nanopockets without and with pretension due to the pressurization of their host drumhead. **d, e** AFM deflection images of ethanol nanopockets without and with pretension, where the images in **d** and **e** are obtained from the same nanopockets with different scanning angle. Scale bars: **b, c** 50 nm, **d, e** 100 nm.

characterization of the wrinkled configurations, we conclude that wrinkles do not behave as channels before the nanopockets merge spontaneously (Supplementary Fig. 14). To figure out the mechanism of these observations, it is essential to first clarify the origin of the driving force for the coalescence of neighboring nanopockets, which may now be understood with the aid of the height profiles.

When the pockets are close, the extended stress/deformation fields outside each nanopocket become interactive, leading to the loss of symmetry. We demonstrate this asymmetry by effective contact angles of 18 neighboring nanopockets and 13 isolated nanopockets with and without pretension. The insets in Fig. 4d, h show the typical height profiles, where the definition of the effective contact angles is also illustrated. For neighboring wrinkled nanopockets, the symmetry is found to be slightly broken, as verified by $\beta_{in} < \beta_{out}$. By contrast, when the host tBLG was stretched by ~1% strain, the difference between the inner-side and outer-side becomes indiscernible (Fig. 4d). Besides, we find that isolated nanopockets (including those formed by nanopocket coalescence) are largely symmetric (Fig. 4h). We also conduct the roundness analysis based on the morphologies of nanopockets to further reveal how the distance between nanopockets and the pretension influence the symmetry of nanopockets (Supplementary Fig. 15). These results, together with the observation of the merging process, suggest a relation between the asymmetric geometry of neighboring nanopockets and the pocket–pocket interactions.

**Long-range interactions**. We illustrate this asymmetry by a 1D cross-section in Fig. 5a. At contact lines of the nanopocket 2, the pretension is superposed by the prescribed $T_{pre}$ and the radial tension owing to the nanopocket 1, $T_{1 \to 2}$. The $T_{1 \to 2}$ decays over the distance $r$ that is defined relative to the center of the

nanopocket 1. This interactive tension decreases as $r^{-2}$ in the absence of the wrinkling (according to the Lamé problem)[45]. However, the formation of the wrinkling would slow the decaying rate from $r^{-2}$ to $r^{-1}$ and thus extends the interacting range. The difference between the pretensions acting on the two "sides" of nanopocket 2 needs to be balanced by the sheet–liquid interaction (this is not true in 1D but should be valid in 3D). We suggest that the net force tends to push the liquid moving toward the coalescence. In addition to this elastic force, the asymmetric pretension could perturb the geometry of the confined liquid, leading to an extra driving force $F_c$ due to capillarity. Specifically, $\beta_L < \beta_R$ in the nanopocket 2 since $T_{1 \to 2}^L > T_{1 \to 2}^R$ (recalling the observation in Fig. 4 and flattening effect of pretension in Fig. 3). Consequently, the net capillary force drives the liquid to the left side as it appears more "hydrophilic". We, therefore, call this elastocapillary mechanism, the essence of which is the deformable geometry and boundaries. Similar concepts have been recently explored for the control of passive motions of micron-sized droplets, including durotaxis[46], tensotaxis[47], and bendotaxis[48].

**MD simulations**. To obtain more insights on the elastocapillarity-driven merging processes of nanoconfined droplets in graphene bilayers, we performed MD simulations (see details in the "Methods" section). Two water-filled nanopockets are intercalated in a graphene bilayer. The snapshots in Fig. 5b show the local morphologies of nanodroplets in the coalescence process. As two nanopockets approach each other, their shapes appear flatter and more asymmetrical, which is consistent with the notion of long-range effective attraction posed in our discussion on the experimental results and theoretical analysis. We find that the coalescence process includes the first stage of attraction ($d > 2.5a$, $a$ is the radius of droplets) driven by the elastic restoration force, and the second step of spontaneous coalescence ($d < 2.5a$) after the initial contact is

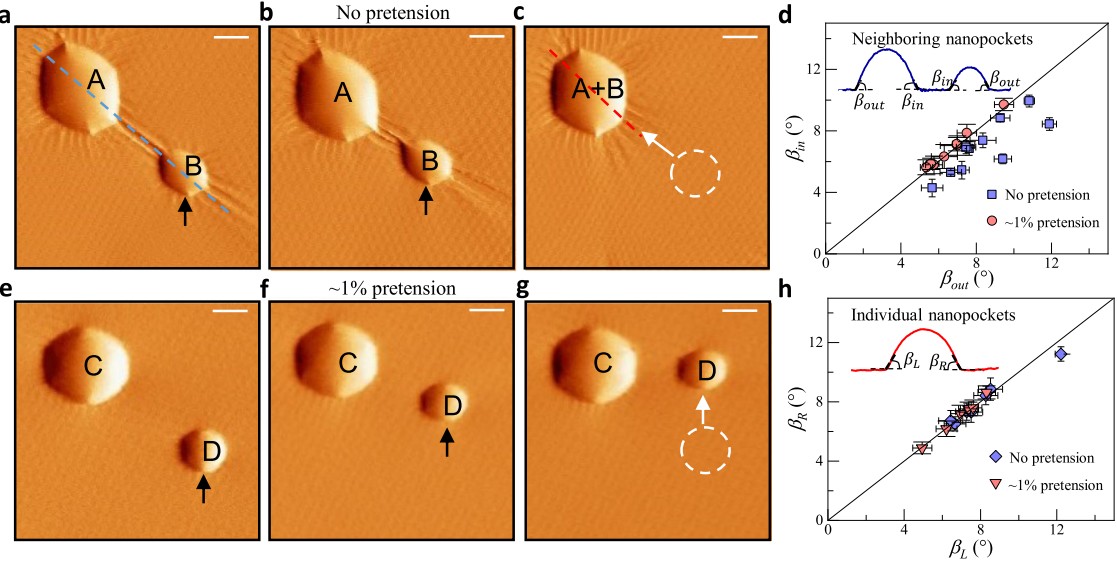

**Fig. 4 Coalescence of nanopockets. a** The initial configurations of nanopockets-A and -B. AFM deflection images of two neighboring nanopockets on tBLG (twist angle $\theta \sim 0.6°$) without pretension, where the AFM tip sweeps on the graphene surface horizontally while moving from the bottom to the top. The black arrows correspond to the direction of the mechanical stimuli. The white dashed circle and the white arrow indicate the initial location and moving direction of nanopocket-B, respectively. The blue dashed line is taken through the centers of the two nanopockets. **b** The configurations of nanopockets -A and -B after the first mechanical stimulus. **c** The configuration of nanopocket-A+B after the second mechanical stimulus. **d** The effective contact angles were measured from the height profiles of 18 neighboring nanopockets with (red circle markers) and without (blue square markers) pretension. The inset shows the height profile of nanopockets-A and -B along the blue dashed line in **a**, where the effective contact angles ($\beta_{out}$ and $\beta_{in}$) are calculated from the linear fitting of the edges of the height profile. The solid line here corresponds to the inner-side and outer-side effective contact angles being equal. For each nanopocket, the contact angles were taken along three directions, from which the average values of $\beta_{out}$ and $\beta_{in}$ as well as x(y)-error bars were determined. **e–g** AFM deflection images of two neighboring nanopockets on tBLG with ~1% pretension, where the implications of the arrows and circle are the same with **a–c**. **h** The effective contact angles ($\beta_L$ and $\beta_R$) measured from the height profiles of 13 single nanopockets with (red triangle markers) and without (blue rhombus markers) pretension. The inset shows the height profile of nanopocket-A+B along the red dashed line in **c**. Again, the solid line corresponds to the left-side and right-side effective contact angles being equal. For each nanopocket, the contact angles were taken along three directions, from which the average values of $\beta_L$ and $\beta_R$ as well as x(y)-error bars were determined. Scale bars: 100 nm.

achieved between the two droplets (Fig. 5d). The potential energy landscape is explored by considering the inter-pocket distance $d$ as an indicator of the coalescence state (Fig. 5c, d). Agreeing with experimental observation in Figs. 3 and 4, radially wrinkling patterns appear around the isolated droplet ($d = \infty$ or 0) in the stress-free graphene bilayer. During the attraction processes (Fig. 5d), a distinct, unidirectional wrinkle nucleates and connects the two droplets (Fig. 5c). However, we note that there are no water channels formed within this wrinkled structure. Such a merging process, therefore, differs from the Ostwald ripening that describes a process of mass diffusion[20].

For nanopockets with pretension, the wrinkling features are weakened according to the criterion in Eq. (4) (wrinkles are indeed absent in the simulations under 2% pretension, Fig. 5c). Aligning with our experimental observation and theoretical prediction, the unidirectional wrinkle connecting the two droplets can be suppressed by pretension. Specifically, the aspect ratio is found to be 0.15 and 0.13 for the system under the pretension of 0% and 2%, respectively. In stretched bilayers, the gradient of potential energy reduction in the attraction stage decreases with the pretension (Fig. 5d). The morphologies of droplets and stress distribution in graphene sheets during coalescence of the nanopockets also indicate that the wrinkled patterns are reduced under pretension (Supplementary Figs. 16–17). These results agree well with the experimental results and the discussion based on the simplified 1D model that the wrinkling can mediate a longer-range attraction between the droplets and this "driving force" is reduced by pretension that suppresses the wrinkles. The second spontaneous stage where the two droplets come into contact and merge is, however, controlled by cohesion between

water molecules in the droplets to a large extent, and is difficult to resolve in experiments.

## Discussion

We have demonstrated the self-cleaning ability of vdW material interfaces, including not only the aggregation of interfacial contamination into nanopockets but also the recovery of interfacial moiré patterns after the motion of these nanopockets. We have investigated the mechanical responses of the nanopockets to the twist and stretch, based on which an elastocapillary mechanism is proposed to explain the driving force for the coalescence. These results could be beneficial to the fabrication of clean vdW material devices using mechanical treatments, as demonstrated in a recent work[49]. Besides, the graphene-confined liquids have found exciting applications in the field of liquid cell EM, where it has been long-desired to in situ image pressure-sensitive processes in materials science, such as colloidal nanocrystal growth and interlayer lithium uptake[27,50]. Our studies advocate that both capillarity and elasticity contribute to the confinement condition (including pressures), opening more possibilities for its deterministic control, for example, via stretching of the graphene sheets.

## Methods

**Fabrication of tBLG-confined water nanopockets.** To fabricate tBLG with encapsulated water molecules, we use a water droplet-assisted method, as illustrated in Supplementary Fig. 1. First, mechanical exfoliation of graphite is employed to prepare graphene monolayer sheets (Supplementary Fig. 1a) on pre-patterned SiO$_2$/Si substrate. A thin polydimethylsiloxane (PDMS) film is placed on a micro-translation stage, which can be precisely controlled under an optical microscope. After dropping deionized water (DI water) on the PDMS surface, we move the stage and approach the target monolayer graphene gently. To ensure the

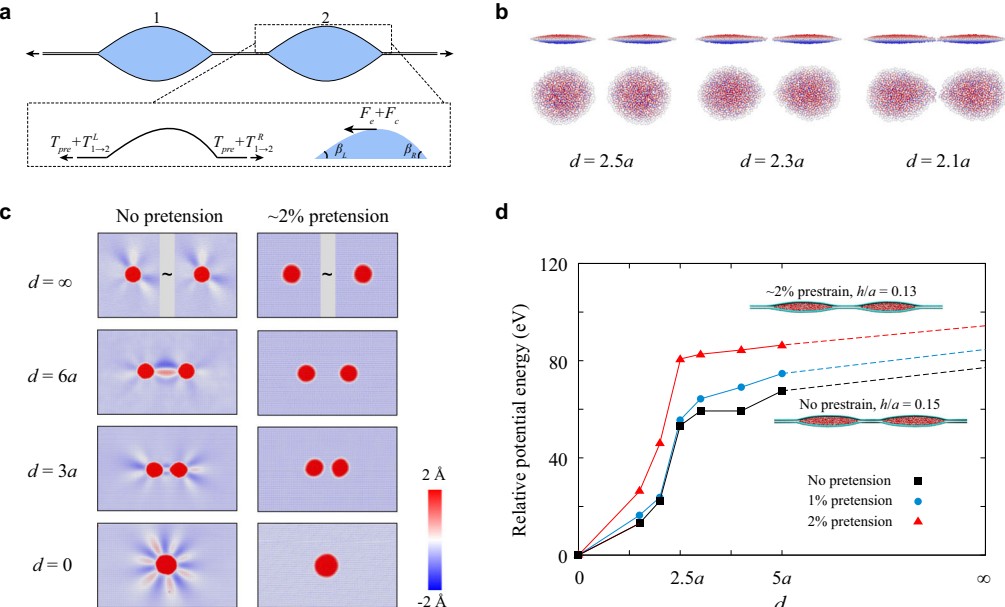

**Fig. 5 Coalescence mechanism. a** Schematic of the driving force for the coalescence process. $T_{pre}$ denotes the pretension applied to both nanopockets. $T_{1\rightarrow2}^{L}$ denotes the tension from nanopocket 1 to the left of nanopocket 2, whereas $T_{1\rightarrow2}^{R}$ denotes the tension from nanopocket 1 to the right of nanopocket 2. $F_e$ represents the elastic force of graphene sheets. $F_c$ represents the capillary force of trapped liquids. $\beta_L$ and $\beta_R$ represent the effective contact angles of the left and right sides of nanopocket 2, respectively. **b** Molecular dynamics simulation snapshots showing the side and top views of droplets during coalescence with a small inter-pocket distance $d$ (compared with the radius of droplets, $a = 4$ nm). The atoms are colored by their displacement out of the bilayer plane. **c** Comparison of the coalescence processes of nanodroplets intercalated in a graphene bilayer with or without pretension. The atoms are colored by their position out of the bilayer plane, where the color range is set to be smaller than the height of blisters to show the fringe of nanopockets and wrinkle patterns. **d** Relative potential energy landscape of the nanopocket-containing bilayers as a function of $d$ under 0, 1, and 2% pretension conditions, where the represented symbols connected with solid lines are simulation measurements, and the dashed lines are energy predictions for larger $d$ according to the trend of solid lines. The simulation snapshots are shown in the inset. The reference potential energy is measured at $d = 0$ (the merged state), which is $-4.678 \times 10^6$, $-4.674 \times 10^6$, and $-4.661 \times 10^6$ eV for 0, 1, and 2% pretension, respectively.

integrity of graphene, the PDMS stamp may contact the edge of the graphene monolayer firstly. After the water droplet on PDMS spreads out on the $SiO_2$ substrate, the PDMS stamp is quickly peeled off by lifting the stage (Supplementary Fig. 1b–d). Next, the PDMS stamp with the picked graphene monolayer is translated to the top of another target monolayer graphene. By rotating the underlying platform with the aid of an optical microscope, we can set the twist angle between the two graphene sheets. When detaching off the PDMS stamp again, the vdW interaction between graphene sheets can keep the tBLG on the substrate (Supplementary Fig. 1e–f). Note that, to make suspended tBLG, we pattern microholes on the supporting $SiO_2$/Si substrate with radii of 1.5 and 2.5 μm as shown in Supplementary Fig. 1d.

**Fabrication of tBLG-confined ethanol nanopockets**. To fabricate tBLG with encapsulated ethanol molecules, we use an ethanol droplet to transfer graphene by the same steps in the liquid-assisted method, where the difference is that water is replaced by ethanol. In the same way, we fabricated the suspended sample by transfer tBLG on a patterned substrate.

**AFM and Raman characterization**. The AFM (Asylum Research Cypher) was used to characterize the morphologies of nanopockets and suspended tBLGs. The mechanical stimulus to nanopockets was applied by the tip that has a normal spring constant of ~0.08 N/m and a torsional spring constant of ~16.0 N/m. The frequency of the probe scanning is fixed to be 6 Hz. A Renishaw system with an incident wavelength of 514 nm was utilized to collect the Raman spectrum. The laser intensity was kept below 0.5 mW to avoid local heating. The spectral resolution was 1.0 $cm^{-1}$, and the spatial resolution was 500 nm.

**Pressurization of the tBLG drumhead**. The suspended tBLGs on the patterned $SiO_2$/Si substrate were placed in a high-pressure autoclave. Then the inert gas ($N_2$) was filled into the autoclave to maintain a constant pressure for several days[39,40]. After the pressure is stabilized, the samples were taken out for characterizations.

**MD simulations**. MD simulations are performed to explore the merging process of water-filled nanopockets intercalated in graphene bilayer, using the large-scale

atomic/molecular massively parallel simulator (LAMMPS)[51]. Two water droplets (~4300 water molecules in each) were intercalated in a graphene bilayer (Fig. 4). A $100 \times 80$ nm$^2$ supercell is constructed with periodic boundary conditions enforced in the in-plane direction and an open boundary in the out-of-plane direction. The adaptive intermolecular reactive empirical bond-order (AIREBO) potential function is used to describe the interatomic interaction in graphene[52], with an adjusted cutoff distance 0.2 nm to avoid the spurious strengthening effect[53]. The rigid SPC/E model is used for water molecules[54], and the SHAKE algorithm is applied for the stretching terms between oxygen and hydrogen atoms to reduce high-frequency vibrations that require a shorter time step for numerical integration, and the long-range Coulomb interactions are computed by using the particle-particle particle-mesh algorithm (PPPM) with an accuracy of $10^{-4}$[55]. The interaction between water and graphene is described by the 12–6 Lennard-Jones potential between oxygen and carbon atoms with parameters $\varepsilon = 4.063$ meV and $\sigma = 0.319$ nm[56].

All equilibration simulations are performed at a constant temperature of 300 K by using the Berendsen thermostat, and the pretension effect is investigated by stretching the simulation box by 1% or 2% along with the in-plane directions. To exclude the fluctuations and provide a direct analysis of the driving force, potential energy evolution during the coalescence is analyzed using constrained energy minimization. The distance between the droplets $d$ (along the zigzag direction in the graphene bilayer) is controlled through their centers of mass, which are tethered to specified positions $R_C$ through an elastic spring with a spring constant $k$. A force of $kR_cM_i/M$ is then applied to each atom, where $M_i$ and $M$ are the masses of the oxygen/hydrogen atom and the nanodroplet, respectively. The shape of nanodroplets is analyzed from 10 ns-equilibration simulations, and the potential energy of the water-bilayer composite is calculated after simulating-annealing processes to 0 K with a cooling rate of $3 \times 10^{-15}$ K/s. For a large value of $k = 1600$ N/m, the fluctuation of $R_c$ is not significant and spontaneous merging processes are observed as the inter-droplet distance decreases to $2.5a$, where $a = 4$ nm is the radius of nanodroplets. For a small value of $k = 0.16$ N/m, however, the fluctuation is significant (±2 nm). The measured potential energy profile is thus noisier, but the main features are preserved. It should be noted that our MD simulations are limited by the size of droplets and graphene bilayer. Large-scale features such as the elliptic shape of nanopockets, which may be attributed to the difference in the bending resistance along with armchair or zigzag directions, cannot be captured. The video of the spontaneous coalescence of nanopockets in

Supplementary Movie 1 shows the simulation trajectories of merging of two water nanodroplets intersected between graphene layers. The initial distance between two nanodroplets is $d = 2.5a$, and the simulation parameters are described in the Methods section. After equilibration of the initial geometry by controlling the centers of mass of each droplet with $k = 1600$ N/m, and under a temperature of 300 K without pretension, water merges spontaneously after the release of the force restriction.

## Data availability

The data that support the findings of this study are available from the corresponding authors on request or in Figshare with the identifier https://doi.org/10.6084/m9.figshare.14909907.

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

## Acknowledgements

This work is jointly supported by the National Natural Science Foundation of China (Grant nos. 11890682, 11832010, 22072031, and 21721002), the Strategic Priority Research Program of Chinese Academy of Sciences (CAS) under the grant nos. XDB36000000, XDB30201000). We thank D. Sanchez for useful edits on an earlier draft. We thank D. Vella for the helpful discussion and comments. Z.D. acknowledges the support of the Graduate Continuing Fellowship from UT Austin.

## Author contributions

Y.H., Z.D., Q.L., L.L., and Z. Z. designed the research; Y.H., S.Z., G.W., and Q.L. carried out the experiments; S.F. and Z.X. contributed to the MD simulations. Z.D. performed the theoretical analysis, Z.D., Y.H., S.F., Z.X., Z.Z, and L.L. wrote the paper. All authors discussed the results and commented on/revised the manuscript.

## Competing interests

The authors declare no competing interests.
