## [Peer Review File · Nature Communications]

Reviewers' Comments:

Reviewer #1:

Remarks to the Author:

The paper describes an interesting experimental work on the coalescence of micro-droplets trapped between a pair of graphene layers. In previous studies, such coalescence process was attributed to Ostwald coarsening which is for instance observed in the condensation of dew. Here, the authors propose a different mechanism based on the squeezing of the droplets between the layers. The authors' argument is relatively simple to understand. When a droplet is trapped, it generates a bump between the layers, which results on radial tension around it. In the case of a single droplet, the strain field is symmetric and the droplet remains in place. However, if two neighboring droplets are trapped the stress field loses its circumferential symmetry, which results in an attractive force between the droplets. In their experiments, the authors are also able to induce a mechanical tension to the membrane, which enhances this effect. In addition to these delicate experiments, the authors propose some modeling with both analytical expressions and molecular dynamics simulations.

I found the experiment intriguing and the proposed mechanism for the coalescence original and convincing. This mechanism seems particularly relevant for low friction systems such as graphene layers. Although the main lines of the theory seem sounded, I am at the moment less convinced by the detail of their derivation.

Nevertheless I think that the originality of the proposed mechanism will motivate further studies not only in the graphene community, but more generally by numerous researchers interested in the mechanics of thin films. I would therefore recommend the publication of an improved version of the manuscript to Nature Communications

I hope the detailed comments below may help bringing some clarifications.

Detailed comments:

line 38-39: "but also raises an ideal platform for the sheets to prevent their functional performance from degradation" -> awkward phrasing

line 43: "substantial enough to migrate these small molecules" -> to drive ?

line 56: "twisted bilayer graphene" -> I would specify for unfamiliar readers that twisted bilayer graphene corresponds to superposed layers with a mismatch of the orientation, which leads to a Moiré pattern.

line 72: although it is probably trivial for specialists, I would add in the supplementary section a sketch to explain the formula.

Figure 1: Fig a is a bit confusing as the reader is first supposed to look at the middle of the image (macroscopic scale) and then at the top and the bottom (microscopic scale).

Caption: "suspending" -> suspended

"with a height of H " -> I would replace by something like "which leads to a shallow spherical shell of height H "

I would specify that that the series b,c, d correspond to a motion induced by the AFM tip, and e-f to the coalescence of a pair of trapped droplets.

line 90: "(or the so-called self-cleaning)" -> the patterns prove that the surfaces are clean, but not that they cleaned themselves. They could be perfectly clean from the beginning. Remove the comment?

line 99: Lotus effect relies on the rolling of water droplets on hydrophobic surfaces to remove contaminants. Do the authors suggest that in the case of graphene, water droplets may be sprayed or condensed at the surface of the layers and then used to remove extra contaminants from graphene?

line 120: "upon a twist" -> I am not sure to get the meaning of this expression.

line 150: Please provide some order of magnitude for E_{2d} , γ_{gg} , γ_{gl} (or of their non-dimensionalized versions).

I have some trouble with Equation 2: If we expand $\cos\theta$, we find $1-2(h/a)^2 +16(h/a)^4$, which then lead to an extra $\tilde{\gamma}_{gl} (h/a)^4$ term. Is $\tilde{\gamma}_{gl}$ much smaller than 1 so that it can be neglected in the equation?

line 157: "follows the same response of the interlayer friction" -> I do not understand the ground for this statement. Is it due to the fact that friction would induce different values for $N+$ and $N-$?

In the discussion of the shape of the bumps, it would be worth mentioning a macroscopic study on "Liquid blister test" Chopin et al., Proc. R. Soc. A 2008 464, 2887-2906

line 180: " $T_{pre} \sim H^2/A^2$ " -> E_{2d} is missing

Equation 3: I have the same issue of the additional term in $(a/h)^4$

Equation 4 says that wrinkles should be always observed in the absence of external stretching of the membrane. Is it always the case experimentally?

line 221: "The formation of the wrinkling slows the decaying from r^{-2} to r^{-1} and thus extends the interacting range." -> I am not convinced that the formation of the wrinkles modify significantly the dissymmetry of the radial strain fields around the droplets.

Figure 4: What is the value selected for a in the MD simulations. This is important to compare with d .

It would be nice if there could be more data points, especially for smaller values of d and to derive a force.

I am also a bit lost, the decrease of energy as the droplets are brought together is more pronounced when there is no prestrain. This seems in contradiction with the general message that prestrain enhances coalescence.

line 304: "pressure of ~ 0 MPa" -> please correct

Supplementary document:

line 48: "arrows" -> lines ?

line 76: I am missing some sketch to understand how the twist angle can be extracted from these oscillations.

Figure S8: I do not see any appreciable difference between the spectra for $\theta \sim 7.3^\circ$ and $\theta \sim 8-9^\circ$. How can the difference of the orientation be inferred?

It would also be nice for a non-specialized reader to mention what G, R and 2D bands correspond to.

line 101: "coalescent" -> coalescence

line 111: "instability free situation" -> situation without wrinkles ?

line 165: "suppress" -> decrease ?

line 176: Why should wrinkles be associated to edge slippage?

Reviewer #2:

Remarks to the Author:

In this paper using AFM technique and molecular dynamics simulations the authors studied the formation of nanobubbles in suspended bilayer graphene which is put on round hole of radius A

made of SiO₂/Si substrate. They characterized the aspect ratio between the height and radius of bubbles and studied the moire superlattice between two graphene sheets outside the bubbles. They also studied self-cleaning and coalescence of neighboring bubbles using AFM tip. Their results suggests that the Ostwald processes is no longer applicable for the coalescence of two bubbles. Here are my comments

1- The nanobubbles have already been observed and characterized by several groups, e.g. Ref. 7, Nature Communications volume 8, Article number: 15844 (2017) and ... Also there are many theoretical study on this topic, e.g. see J. Appl. Phys. 112, 083512–083519 (2012).

2- The moire bands in twisted double-layer also studied by several authors e.g. see PNAS July 26, 2011 108 (30) 12233-12237 and PNAS 2017 114 (13) 3364-3369, Phys. Rev. Lett. 106, 126802 (2011) and several other papers.

3-How the reported scaling in Fig. 2(a) can be related to that of reported in Fig. 2(a) of ref. 7? How the presence of different contaminants can change the scaling reported here.

4-I am not sure that the dynamics and merging of the nanobubbles were studied carefully in this study. There are a lot of theoretical estimation but not satisfactory. For example the time scale for merging in experiment cant be comparable with those reported in MD simulations. Using MD technique one is able to straightforwardly characterize merging phenomena, but what can be interesting is to provide more experimental data. There are a lot of discussion in the supplementary about pretension but no clear relation with experiment and theory. One issue that could be interesting to look, is measuring the phase state of trapped water and strain on the bubble.

It would also be interesting to see is there self-cleaning property for other kind of contaminant, such as Benzene, Hexane and ethanol? This should be done in order to make the self cleaning idea complete/solid.

I dont think the current provided information for graphene nanobubbles is enough to being publishable in nature communications.

Reviewer #3:

Remarks to the Author:

The authors explore nano-pockets of water and impurities between 2 graphene sheets using atomic force microscopy. They theoretically model the geometry of the pockets and discuss the roles of wrinkles ("instabilities") in the coalescence process of the bubbles.

I do not recommend this paper for publication in Nature Communication as it lacks in sufficient novelty, it lacks a clear message about the relevance of the results, and contains several false claims of novelty. The part of the conclusions that are supported by data reveal only incremental progress, and are better suited for more specific journal.

The novel part of the work is the (anecdotal) observation of the wrinkles around nanopockets, exploration of the effect of external tension and bilayer twist angle on the geometry of the nanopockets (no significant effect), and extension of previously published theoretical model to account for these results. All those results are either not explored sufficiently and not supported by data (wrinkle effects not quantified, just noted), or they offer only incremental insight.

Allow me to dissect the paper:

1. Authors transferred 2 layers of graphene over hole that lead to nanopockets (not new, observed

previously many times, see Ref-18 and Ref-A)

2. Authors claim this is the first report of AFM observation of Moiré patterns in suspended bilayer graphene, line# 77-79 (false claim, previously published by authors themselves and not cited – Ref-A)
3. Authors observed coalescence of nanopockets with contact AFM, they also claim it could be driven by AFM tip – not new, but authors spend lots of time discussing it, lines#89-110
4. Authors elaborate a model in details to describe the shape of the bubble. Not new, not properly cited, misrepresented – this model is the from paper of one of the authors, Ref 8 – paper cited in general context at the begging, but not as a source of the model. lines#120-153
5. Authors theoretically correlated graphene-graphene interface energy as a function of twist angle (novel, but it is incremental improvement on the above model).
6. Experimentally it is impossible to see if the above point holds. (Figure 2 is hard to read, but does not explicitly reveal any trend in h/a vs. twist angle) Even if it shows some trend, it would not be surprising, since the effect of twist angle on interfacial energy has been observed before (Ref 31, 32) lines#150-158
7. Authors apply external pressure on graphene membrane (not new), and observe effect of external tension on the shape of nanopockets (new – both experimentally and theoretically, but not super-exciting) lines #175-192
8. Authors explore experimentally wrinkling around nanopockets (new and interesting, but anecdotal) They show several images, and of wrinkling (and lack of), but they do not quantify the effect. It is not clear if this effect is artefact of AFM imaging or not.
9. They postulate the wrinkling is due to strain field around the bubble, which leads to long-range nanopocket interaction and their coalescence. The claim is not supported by data. No clear trend of wrinkling vs. coalescence is established (Fig3b,c have opposite trends than expected)
10. They theoretically correlated appearance of instabilities (wrinkling) with geometry of nanopockets and external strain (novel), but there is no proper quantitative comparison with experiments (because experiments are not properly quantified) Some observations are off predictions, but they are waved away by hidden variables. Lines# 197-202
11. There is no discussion how MD simulations match theoretical considerations lines#238-256. The results (Fig4c) seem in contradictions with the theoretically predicted scaling (lines 219-224)
12. The claims in the title and abstract should be focused in graphene, not general “van der Waals materials”. Some of the effect, including wrinkling and lubrication, might not generalize to the full extent in the other 2D materials (improper generalization)

One concerning issue is that the paper has 4 corresponding authors and 3 equal-contribution authors, while the volume of work is not much different that in most of the papers. This represents a strong inflation of the authorship contributions, and should have a high bar of justification.

Extra references

A) Hou, Y. et al. Preparation of Twisted Bilayer Graphene via the Wetting Transfer Method. ACS Appl. Mater. Interfaces 12, 40958–40967 (2020)

Reviewers' comments:

Reviewer #1:

The paper describes an interesting experimental work on the coalescence of micro-droplets trapped between a pair of graphene layers. In previous studies, such coalescence process was attributed to Ostwald coarsening which is for instance observed in the condensation of dew.

Here, the authors propose a different mechanism based on the squeezing of the droplets between the layers. The authors' argument is relatively simple to understand. When a droplet is trapped, it generates a bump between the layers, which results on radial tension around it. In the case of a single droplet, the strain field is symmetric and the droplet remains in place. However, if two neighboring droplets are trapped the stress field loses its circumferential symmetry, which results in an attractive force between the droplets. In their experiments, the authors are also able to induce a mechanical tension to the membrane, which enhances this effect.

In addition to these delicate experiments, the authors propose some modeling with both analytical expressions and molecular dynamics simulations.

I found the experiment intriguing and the proposed mechanism for the coalescence original and convincing. This mechanism seems particularly relevant for low friction systems such as graphene layers. Although the main lines of the theory seem sounded, I am at the moment less convinced by the detail of their derivation. Nevertheless, I think that the originality of the proposed mechanism will motivate further studies not only in the graphene community, but more generally by numerous researchers interested in the mechanics of thin films. I would therefore recommend the publication of an improved version of the manuscript to Nature Communications.

I hope the detailed comments below may help bringing some clarifications.

Our response: We thank the reviewer for this enthusiastic assessment and positive comments on our work. These comments are detailed and helpful to improve the presentation of our manuscript.

The reviewer has the following detailed comments (which we appreciate very much):

1) line 38-39: “but also raises an ideal platform for the sheets to prevent their functional performance from degradation” -> awkward phrasing

Our response: We have modified to be “but also prevent their functional performances from degradation”. (Lines: 40-41, Manuscript)

2) line 43: “substantial enough to migrate these small molecules” -> to drive ?

Our response: Thanks for the suggestion. We have changed this accordingly. (Line: 44, Manuscript)

3) line 56: “twisted bilayer graphene” -> I would specify for unfamiliar readers that twisted bilayer graphene corresponds to superposed layers with a mismatch of the orientation, which leads to a Moiré pattern.

Our response: Thanks for the suggestion. We have added the description in the main text: “we focus on one of the most promising vdW materials – twisted bilayer graphene (tBLG) that corresponds to superposed

layers with a mismatch of the orientation, which leads to the formation of moiré patterns”. (Lines: 57-59, Manuscript)

4) line 72: although it is probably trivial for specialists, I would add in the supplementary section a sketch to explain the formula.

Our response: We provide a sketch in Fig. S7 in the supplementary section as well as the reference for the deviation (also see Fig. R1 below). (Lines 80-87, SI)

Fig. R1. (a) Sketch of moiré patterns in which θ is the twist angle and L is the length for the periodic pattern [Kuwabara, M., et al., *Appl. Phys. Lett.* **56**(24): 2396-2398 (1990)]. (b) Lateral force AFM images of moiré pattern. The twist angles of samples are $\sim 0.6^\circ$, $\sim 1.5^\circ$, $\sim 7.3^\circ$, which are determined according to the periodicity shown in (c). Scale bars: (b) 20 nm.

5) Figure 1: Fig a is a bit confusing as the reader is first supposed to look at the middle of the image (macroscopic scale) and then at the top and the bottom (microscopic scale).
Caption: “suspending” -> suspended

Our response: We have modified the figure and the caption according to these suggestions. The revised version has been listed as Fig. R2. (Lines: 81,82, Manuscript)

Fig. R2. The revised schematic in Figure 1a (in Manuscript).

6) “with a height of H” -> I would replace by something like “which leads to a shallow spherical shell of height H”

Our response: Thanks for this suggestion. We have adopted it in our revised manuscript. The caption in Fig. 1 has been changed as “The tBLG can be pressurized by gas which leads to a shallow spherical shell of height H ”. (Lines: 83,84, Manuscript)

7) I would specify that the series b, c, d correspond to a motion induced by the AFM probe, and e-f to the coalescence of a pair of trapped droplets.

Our response: We have specified (b-d) and (e, f) in the caption of Figure 1 accordingly. (Lines: 85-87, Manuscript)

8) line 90: “(or the so-called self-cleaning)” -> the patterns prove that the surfaces are clean, but not that they cleaned themselves. They could be perfectly clean from the beginning. Remove the comment?

Our response: We agree and have removed this comment. (Line: 93, Manuscript)

9) line 99: Lotus effect relies on the rolling of water droplets on hydrophobic surfaces to remove contaminants. Do the authors suggest that in the case of graphene, water droplets may be sprayed or condensed at the surface of the layers and then used to remove extra contaminants from graphene?

Our response: We didn't suggest this. Apparently, we don't have enough experimental evidence to recall the lotus effect, so we decided to remove this in the revised manuscript. (Line: 98, Manuscript)

10) line 120: “upon a twist” -> I am not sure to get the meaning of this expression.

Our response: To clarify, we used “Geometry of nanopockets in tBLG with various twist angles” instead. (Line 118, Manuscript)

11) line 150: Please provide some order of magnitude for E_{2D} , γ_{gg} , γ_{gl} (or of their non-dimensionalized versions).

Our response: E_{2D} of graphene has been thought to be ~ 340 N/m and the typical value for these interfacial energies should be in the order of 0.1 J/m². We provide the order of magnitude of the non-dimensionalized interfacial energies in the revised manuscript, which is $\sim 10^{-4}$. (Lines: 150-152, Manuscript)

12) I have some trouble with Equation 2: If we expand $\cos\theta$, we find $1-2(h/a)^2 +16(h/a)^4$, which then lead to an extra $\tilde{\gamma}_{gl} (h/a)^4$ term. Is $\tilde{\gamma}_{gl}$ much smaller than 1 so that it can be neglected in the equation?

Our response: Yes, the $\tilde{\gamma}_{gl}$ should be $\sim 10^{-4}$. (Line: 152, Manuscript)

13) line 157: “follows the same response of the interlayer friction” -> I do not understand the ground for this statement. Is it due to the fact that friction would induce different values for N^+ and N^- ?

Our response: We apologize for the confusion this statement may bring. What we meant by this is that the interlayer friction decreases as the twist angle increases. What we wanted to convey is that overall, the

adhesion between graphene sheets seems to decrease as the twist angle increases. In this regard, there is some similarity between frictional and adhesive behavior. However, to avoid any confusion, we decided to remove this statement in our revised manuscript. (Line: 153, Manuscript)

14) In the discussion of the shape of the bumps, it would be worth mentioning a macroscopic study on “Liquid blister test” Chopin et al., Proc. R. Soc. A 2008 464, 2887-2906

Our response: Yes, it is very worth mentioning this relevant reference there though we have mentioned this paper in the introduction when talking about the liquid-thin film interactions. (Line: 143, Ref. 35, Manuscript)

15) line 180: “ $T_{pre} \sim H^2/A^2$ ” -> E_{2d} is missing

16) Equation 3: I have the same issue of the additional term in $(a/h)^4$

Our response: Thanks for spotting this typo. We have corrected it in our revised manuscript. (Lines: 175, 187, Manuscript)

17) Equation 4 says that wrinkles should be always observed in the absence of external stretching of the membrane. Is it always the case experimentally?

Our response: The short answer may be Yes. But in experiments, with zero applied stretching, we found the wrinkles show up around the nanopocket, (see Fig. R3, also Supplementary Figure 11).

Fig. R3. Water and ethanol nanopockets encapsulated at bilayer graphene interfaces. Scale bars: 100 nm.

18) line 221: “The formation of the wrinkling slows the decaying from r^{-2} to r^{-1} and thus extends the interacting range.” -> I am not convinced that the formation of the wrinkles modify significantly the dissymmetry of the radial strain fields around the droplets.

Our response: We did not perform 3D numerical calculations to address this concern properly. But in the revised manuscript, we show some experimental results in Fig. 4c in which asymmetry seems to be more significant in wrinkled samples than in unwrinkled samples. (Line: 219, Manuscript)

19) Figure 4 (now Figure 5): What is the value selected for a in the MD simulations. This is important to compare with d . It would be nice if there could be more data points, especially for smaller values of d and to derive a force.

Our response: In MD simulation, the radius of nanopocket is set to is ~ 6 nm. To clarify our simulation results, the expressions of distances between neighboring nanopockets have been changed into “ $d = na$ ”, as shown in Fig. 5 in the manuscript. Besides, additional simulations were performed for smaller distances: $d = 2.6a$, $d = 2.4a$, $d = 2.1a$. Fig. R4 shows the morphology evolution of nanopockets and trapped droplets at different distances. As the distance between nanopockets decreases, we find that the droplets become largely asymmetric (the flatter edge appears in the inner sides). Due to the limitation of our MD model, we can only estimate the potential energy while not the force in the coalescence process. But we think the driving force is another interesting problem to study in our follow-up work. (Line: 260, Manuscript)

Fig. R4 Morphology evolution of nanopockets (a-c) and trapped droplets (d-i) at different distances.

20) I am also a bit lost, the decrease of energy as the droplets are brought together is more pronounced when there is no prestrain. This seems in contradiction with the general message that prestrain enhances coalescence.

Our response: Many thanks for this comment. Intuitively, prestrain facilitates coalescence. But what we found in simulations is that this intuition is only true when the nanopockets are very close to one another. In the case that neighboring nanopockets with relatively large distance, the asymmetry due to the extended stress field outside the nanopockets is important for the occurrence of coalescence. The asymmetry is suppressed when the uniform pretension is applied to the pockets. This may result from that the asymmetric stress due to the pocket-pocket interaction appears to be relatively less significant when there is a global, uniform pretension. This is interesting, and the experimental results in Fig. 4 may partially show such a mechanism. (Lines: 219-260, Manuscript)

21) line 304: “pressure of ~ 0 MPa” -> please correct

Our response: Thanks for spotting this. We have corrected it in the revised manuscript. (Line: 358, Manuscript)

22) Supplementary document:
line 48: “arrows” -> lines ?

Our response: Thanks for spotting this. We have corrected it in the revised supplementary document. (Line: 53, SI)

23) line 76: I am missing some sketch to understand how the twist angle can be extracted from these oscillations.

Our response: We have added the sketch in Fig. S7 in SI. In detail, we calculate the period of moiré pattern L from the oscillations. Then we can extract the twist angle θ by plugging the period into the formula, $\theta = 2\arcsin(2a/L)$, where a is the graphene lattice constant. (Lines: 83-87, SI)

24) Figure S8: I do not see any appreciable difference between the spectra for $\theta \sim 7.3^\circ$ and $\theta \sim 8-9^\circ$. How can the difference of the orientation be inferred? It would also be nice for a non-specialized reader to mention what G, R and 2D bands correspond to.

Our response: For bilayer graphene with twist angle $\sim 7.3^\circ$ and $\sim 8-9^\circ$, the subtle difference in Raman spectra is the peak position of R' band. Specifically, the peak position of R' band located $\sim 1620 \text{ cm}^{-1}$, $\sim 1627 \text{ cm}^{-1}$ for $\sim 7.3^\circ$ and $\sim 8^\circ$ tBLG, respectively. This agrees with the DFT calculation in Carozo, V., et al., *Nano Lett.* **11**, 4527-4534 (2011).

Thanks for the suggestion. To clarify, we have added the following information in the caption of Supplementary Figure 8: “Note that the Raman G band that locates at $\sim 1580 \text{ cm}^{-1}$, originates from a conventional first-order Raman scattering process and corresponds to the in-plane, zone center, doubly degenerate phonon mode (transverse (TO) and longitudinal (LO) optical) with E_{2g} symmetry [Tuinstra, F., et al., *J. Chem. Phys.* **53**, 1126–1130 (1970)]. While the 2D band (also called G' band) is associated with a second-order process, involving two iTO phonons near the K point for the 2D band [Maultzsch, J., et al., *Phys. Rev. B*, **70**, 155403 (2004)]. The R and R' band appears only in tBLG, since these two bands come from the graphene moiré superlattice [Carozo, V., et al., *Nano Lett.* **11**, 4527-4534 (2011)].” (Lines: 89-100, SI)

25) line 101: “coalescent” -> coalescence

Our response: Many thanks for spotting this. We have corrected the error. (Line: 109, SI)

26) line 111: “instability free situation” -> situation without wrinkles ?

27) line 165: “suppress” -> decrease ?

Our response: Yes, these suggestions do sound better. We have revised the manuscript as your suggestions. (Lines: 155, 209, SI)

28) line 176: Why should wrinkles be associated to edge slippage?

Our response: It happened in our modeling that the graphene-graphene interface is very slippery, so that we do not consider frictional force (if any). Instead of edge slippage, it is more appropriate to use radial displacement at the edge (we have corrected in the revised manuscript and supplementary document), which could cause wrinkling when the displacement is negative (i.e., radially inward). (Line: 220, SI)

Reviewer #2:

In this paper using AFM technique and molecular dynamics simulations the authors studied the formation of nanobubbles in suspended bilayer graphene which is put on round hole of radius A made of SiO_2/Si substrate. They characterized the aspect ratio between the height and radius of bubbles and studied the moiré superlattice between two graphene sheets outside the bubbles. They also studied self-cleaning and coalescence of neighboring bubbles using AFM probe. Their results suggest that the Ostwald processes is no longer applicable for the coalescence of two bubbles. Here are my comments

Our response: We thank the reviewer for reviewing our manuscript and providing the following specific comments that have been very helpful.

1) The nanobubbles have already been observed and characterized by several groups, e.g. Ref. 7, Nature Communications volume 8, Article number: 15844 (2017) and ...

Also there are many theoretical study on this topic, e.g. see J. Appl. Phys. 112, 083512–083519 (2012).

Our response: We thank the reviewer for pointing out these references that have encouraged us further clarify the distinctive contribution of our work. We think we have addressed a novel equation as to why and how droplets confined by two atomic sheets may merge spontaneously—a process of ripening. A particular difference between this work and previous elegant works such as Ref. 7, Nature Communications 8, 15844 (2017) comes from the highly deformable boundary of the nanobubbles we considered here. The deformed boundary not only allows interesting wrinkling patterns surrounding bubbles but also enables fairly effective bubble-bubble interactions, which could further make self-cleaning possible. Experimentally, the new element this work added is the demonstration of using mechanical stretch to control the formation of wrinkling patterns. Theoretically, though the picture looks simple, it is a nontrivial task to understand the complex coupling of elastic instabilities, surface forces, and elastic forces in our system. In this work, we investigate these subtleties via a combination of nanoscale experiments and continuum theories. The reference [J. Appl. Phys. 112, 083512–083519 (2012)] has been cited as Ref. 31 in Manuscript, Line: 121.

2) The moiré bands in twisted double-layer also studied by several authors e.g. see PNAS July 26, 2011 108 (30) 12233-12237 and PNAS 2017 114 (13) 3364-3369, Phys. Rev. Lett. 106, 126802 (2011) and several other papers.

Our response: We thank the reviewer for reminding these important references, which enlighten us to further illustrate the uniqueness of research about moiré pattern in our revised manuscript. People have carried out a series of in-depth studies for moiré bands in twisted double-layer graphene, especially for the changes in energy band and electronic structure caused by moiré superlattice.

Instead of focusing on the physical properties of moiré bands as most of the previous works are, we used

them as a tool to help us better understand the mechanics of van der Waals (vdW) interfaces, in particular, how vulnerable to adventitious contaminants the moiré bands or the graphene interfaces are. We think it is an interesting property of vdW interfaces to be able to recover from contamination, as demonstrated in our work. It is meaningful to understand what causes such intriguing ability, for which we proposed and tested a novel elastocapillary mechanism that is essentially different from the traditional Oswald ripening mechanism. The references [PNAS, 2011, 26, 108(30), 12233-12237; PNAS, 2017, 114(13), 3364-3369; Phys. Rev. Lett., 2011, 106, 126802] have been mentioned as Ref. 4, 21, 22 in Manuscript (Lines: 40, 59).

3) How the reported scaling in Fig. 2(a) can be related to that of reported in Fig. 2(a) of ref. 7? How the presence of different contaminants can change the scaling reported here.

Our response: We thank the reviewer for this helpful comment. The scaling in Fig. 2a can be related to that of reported in Fig. 2a of ref. 7 in the sense that both of them compare the interfacial energies with the elastic energies stored in 2D materials and obtain the ratio of height and radius of nanobubbles as a characteristic parameter. But a quantitative comparison between the two is challenging since the configuration is slightly different.

Following the reviewer's suggestion, we have performed experiments on ethanol contaminants. Fig. R5 shows the aspect ratios of water and ethanol nanopockets under the same twist angle of $\sim 0.6^\circ$ (also see Supplementary Figure 11). We find that the aspect ratios of ethanol nanopockets are ~ 0.106 , which is ~ 0.108 for water nanopockets. The reason for this may be that the graphene-graphene interfacial energy may dominate over the graphene-water or graphene-ethanol interfacial energy. We have dedicated a paragraph in the main text (Lines: 207-216) and a new section in the supplementary information to discuss the results of ethanol nanopockets (Lines: 112-135, SI).

Fig R5. Comparison between water and ethanol nanopockets. AFM images showing the wrinkles around water (a) and ethanol nanopockets (b). (c) Aspect ratios of water and ethanol nanopockets. Scale bars: 100 nm.

4) I am not sure that the dynamics and merging of the nanobubbles were studied carefully in this study. There are a lot of theoretical estimation but not satisfactory. For example, the time scale for merging in experiment can not be comparable with those reported in MD simulations.

Our response: We are grateful to the reviewer for this opinion to help us clarify our work. In the manuscript, we have shown the lateral force images of nanopockets before and after merging in Fig. 1e and f. Although the merging process is difficult to be fully characterized by AFM, we still try to capture the path of the

moving process of nanopockets. As indicated by the blue square in Fig. R6, we used the AFM probe to scan the edge of the nanopocket to provide the force stimuli. Figure R6b shows that the nanopocket remains in place at first, while the probe scanned in the middle, the signal suddenly flattens, indicating that the nanopocket escaped. In other words, the movement of the nanopocket occurs during one turn of the AFM probe scanning. The frequency of the probe scanning is fixed to be 6 HZ, so we can estimate that the movement of the nanopockets happens much less than 0.15 s (Line: 339, Manuscript). We should note that the time scale for merging in MD simulation is extremely small (~ 10 ns), which is difficult to be compared or measured in real experiments. But it may still be inferred that the movement or merging of the nanopockets is a fast process, and the recovering of the van der Waals interface also happens in a short time. Furthermore, the path of moving or merging of nanopockets is studied in experiments to reveal more dynamic behaviors, which we summarized in our response to the next comment.

Fig. R6 (a) AFM lateral force image of neighboring nanopockets. The blue square and arrow indicate the area and direction of the mechanical stimulus, respectively. (b) Image of movement of nanobubble where the AFM tip sweeps on the graphene surface horizontally while moving from the bottom to the top. (c) AFM lateral force image of nanobubble after the merging process. Scale bars: (a)(c) 30 nm, (b) 10 nm.

5) Using MD technique one is able to straightforwardly characterize merging phenomena, but what can be interesting is to provide more experimental data. There are a lot of discussion in the supplementary about pretension but no clear relation with experiment and theory.

Our response: We thank the reviewer for pointing out the potential interests for more experimental evidence. We agree with the reviewer that further experimental support on the dynamics and merging behaviors of nanopockets would be interesting. We, therefore, made the following efforts in our revised manuscript and supplementary information, as summarized below:

Merging phenomena: We first fabricate the suspended ethanol nanobubble without pretension and record the initial morphologies of nanopockets. Figure R7 shows a region containing four nanopockets. Similar to what we discussed before about water nanopockets, we find wrinkles around relatively isolated nanopockets (for instance, nanopocket-D in Fig. R7) as well as wrinkling “connection” between adjacent nanopockets (such as A, B, and C). To provide further insights into the merging phenomena, we use the AFM probe to apply the mechanical stimulus to the nanopocket-B (again, the probe scanning direction is from bottom to top). Though the force applied by the probe is alternating, we find the nanopocket-B move along with the wrinkles between the nanopocket-A and -B as shown in Fig. R7b. When being scanned again, the nanopocket-B merges with the nanobubble-A to form nanopocket-A+B. Meanwhile, Fig. R6c shows that the wrinkles between the nanopocket-B and nanopocket-C disappear with the recovery of uniform moiré patterns. Alternatively, we perform the same scanning on the nanopocket-C, which is relatively far away

from nanopocket-A. Figures R7d and e show that the nanopocket-C moves along with the probe scanning direction (from bottom to top), but not directly towards the nanopocket-A+B. This new result is to support one of the key messages of our work that the nanopocket-nanopocket interaction exists between the neighboring nanopockets, which is particularly significant for those that are connected by wrinkling patterns. These discussions have been added in the revised version of our work (specifically, Lines: 219-233 in Manuscript; Lines: 130-135 in SI)

Fig. R7 (a-e) Movement and merging of nanopockets after mechanical stimuli without pretension. The white arrows correspond to the directions of mechanical stimuli, while the black arrows correspond to the movement of nanobubbles. Scale bars: 50 nm.

Pretension effect on the wrinkling geometry. In the revised manuscript, combining theory and experiments we have discussed the nanopocket geometry in a more quantitative way. As before, we firstly focus on relatively isolated nanopockets, in particular, under which pretension the wrinkling would be suppressed. Experimentally, we are able to use AFM images to identify whether the wrinkling exists for various applied pretension (Fig. R8). Theoretically, our model predicts a simple geometrical criterion for wrinkles being suppressed by $\left(\frac{h}{a}\right)^2 \leq 1.8\left(\frac{H}{A}\right)^2$, which is not dependent on the any specific substance of the trapped liquid within the nanopocket. The comparison between experimental observations and theoretical prediction has been provided in Fig. 3a and discussed in the main text (Lines: 180-198) and SI (Lines: 120-127).

Fig. R8 (a-c) Wrinkling geometry around nanopocket subject to pretension. The white dashed circles are used to show the wrinkled regions (if any). The pretension was indicated by aspect ratios (H/A) of the host tBLG. Scale bars: 50 nm.

Pretension effect on the merging phenomena. When the pretension on nanopockets increases, it is found that the wrinkled region appearing at the pretension-free state starts to shrink and finally disappears. We further investigate the merging of pretensioned nanopockets whose wrinkles have been suppressed (Fig. R9, also Fig. 4 in the main text). We show in Fig. R9 that the nanopocket-F moves along the direction of probe scanning rather than toward the nanopocket-E. This observation actually agrees with one of our previous theoretical conclusions that the pretension suppresses the formation of wrinkling so that the pocket-pocket interaction is somewhat degraded. Since the asymmetry of the nanopocket when interacting with another provides the driving force for the merging, we further provide the inner and outer angles of the asymmetric

nanopockets in Fig. 4c to reveal how the pretension affects the degree of the asymmetry. These results have been discussed in the revised manuscript (Lines: 235-258).

Fig. R9 (a-c) Movement of nanopockets subject to pretension. The white arrows correspond to the directions of mechanical stimuli, and the white dashed circle corresponds to the initial position of the nanobubble. Scale bars: 50 nm.

6) One issue that could be interesting to look, is measuring the phase state of trapped water and strain on the bubble.

Our response: We thank the reviewer for this helpful comment. We agree with the review that the phase state of trapped water is an important issue that has attracted much attention. For example, graphene has been used to trap water molecules to observe the atomic structure of 2D ice [Algara-Siller, G., et al., *Nature* **519**(7544), 443-445 (2015)]. However, in our experiments, due to the limitation of AFM for characterizing the structural information of the internal liquid, it is difficult to reveal the phase state of water at present precisely. We tried to capture moiré patterns between graphene and 2D ice, but the complete results have not yet been formed and would be a bit out of the focus of the current work. So we think it is more appropriate to discuss the phase state of water separately in follow-up work.

Using AFM to measure the deformation of graphene lattice, we were able to roughly estimate the strain on the nanopocket. As shown in Figs. R10b and c, the images of graphene lattice in different regions are obtained by FFT transformation of lateral force images. Based on the scanning lines in Figs. R10b and c, the mean spacings of the graphene lattices were calculated as 0.2467 nm and 0.2497 nm, which amount to 0.28% and 1.51% strain, respectively. These results are in the same order as our theoretical results $\sim \mathcal{O}(1\%)$ but a high-precision conclusion is somewhat limited due to the thermal drift effect of AFM-based measurements.

Fig. R10 (a) AFM height image of a graphene nanopocket. (b) Atomic image of the edge of nanopocket as indicated by the yellow square in (a). (c) Atomic image of the center of nanopocket as indicated by the blue square in (a). (d) Comparison of graphene lattice constants by scanning lines from (b) and (c). Scales bar: (a) 50 nm, (b)(c) 0.2 nm.

7) It would also be interesting to see is there self-cleaning property for other kind of contaminant, such as Benzene, Hexane and ethanol? This should be done in order to make the self-cleaning idea complete/solid.

Our response: We appreciate the reviewer for this idea, which we also think is interesting and would provide complementary support of the self-cleaning phenomenon. For safety reasons, we only conducted a series of experiments using ethanol (see experimental method). Briefly, we use the ethanol droplet to transfer graphene to fabricate the suspended twisted bilayer graphene, following the same steps in section 1 in SI. We found that, like water molecules, the inserted ethanol molecules can also be spontaneously squeezed into nanopockets. Moreover, the movement and merging of ethanol nanobubbles could also be stimulated by probe scanning (as we showed in Figs. R6 and R7). The recovery of the moiré pattern is also observed to support the robustness of the self-cleaning phenomenon. These results have been discussed in the main text (Lines: 219-247, Manuscript).

8) I don't think the current provided information for graphene nanobubbles is enough to being publishable in nature communications.

Our response: We thank the reviewer for these thoughtful comments. We have considered them in our revised manuscript and SI and found them to be very helpful to improve the quality of our work. We hope that our revisions make the manuscript suitable for publication in Nature Communications.

Reviewer #3:

The authors explore nano-pockets of water and impurities between 2 graphene sheets using atomic force microscopy. They theoretically model the geometry of the pockets and discuss the roles of wrinkles (“instabilities”) in the coalescence process of the bubbles.

Our response: We thank the reviewer for reviewing and commenting on our manuscript.

I do not recommend this paper for publication in Nature Communication as it lacks in sufficient novelty, it lacks a clear message about the relevance of the results, and contains several false claims of novelty. The part of the conclusions that are supported by data reveal only incremental progress, and are better suited for more specific journal.

Our response: We regret that the earlier version of the manuscript did not convey the findings clearly, which has led to a number of concerns regarding the novelty and relevance. We have revised the manuscript by modifying the presentation and including a few new experiments and discussions along with theoretical results. In the following, we present point-to-point responses to the reviewer’s comments, which we hope to clarify the novelty and relevance of our work and make the manuscript suitable for publication in Nature Communications.

The novel part of the work is the (anecdotal) observation of the wrinkles around nanopockets, exploration of the effect of external tension and bilayer twist angle on the geometry of the nanopockets (no significant effect), and extension of previously published theoretical model to account for these results. All those results are either not explored sufficiently and not supported by data (wrinkle effects not quantified, just noted), or they offer only incremental insight.

Our response: We thank the reviewer for commenting on our work and letting us know your concerns. We would like to argue the following points that we feel important, or we have made changes/additions according to the reviewer’s specific comments later on.

Novelty. We think we have addressed an important equation as to why and how droplets confined by two atomic sheets may merge spontaneously – a process of ripening. Previous understanding has been based on the Ostwald ripening mechanism, but the role of elastic deformation of the thin sheets has not been previously appreciated. The new mechanism we present here considers both elasticity of the thin sheet and the interfaces of the system would better reveal the underlying physics of this important ripening/merging process.

Pretention effect. We appreciate this comment as it, together with the comments from other reviewers, has motivated us to perform supplemental experiments and analysis in efforts to consolidate our findings. We have explored the effect of external tension and twist angle on the nanopocket geometry. Though the angle only affects the aspect ratios of these nanopockets, external tension plays a key role in suppressing the formation of the wrinkling pattern, which is crucial for the merging process. We justify this statement by the following discussions based on newly added experiments and analysis. These discussions could also be found in the manuscript (Lines: 118-159; 160-198) and SI (Lines: 120-127).

Added discussions 1: Suppression of wrinkling patterns. We agree with the reviewer that the earlier version of our work has not sufficiently discussed the instability criterion. In the revised version of our work, we have included experimental data about the wrinkling patterns more quantitatively, in particular, under which

condition pretension the wrinkling would be suppressed. Experimentally, we are able to use AFM images to identify whether the wrinkling exists for various applied pretension (Fig. R11 below). Theoretically, our model predicts a simple geometrical criterion for wrinkles being suppressed by $\left(\frac{h}{a}\right)^2 \geq 1.8 \left(\frac{H}{A}\right)^2$, which is not dependent on any specific substance of the trapped liquid within the nanopocket. The comparison between experimental observations and theoretical prediction has been provided in Fig. 3a and discussed in the main text (Lines: 180-198) and SI (Lines: 120-127).

Fig. R11 (a-c) Wrinkling geometry around nanopocket subject to pretension. The white dashed circles are used to show the wrinkled regions (if any). The pretension was indicated by aspect ratios (H/A) of the host tBLG. Scale bars: 50 nm.

Fig R12. The coalescence of neighboring nanobubbles. The white arrows indicate the direction of mechanical stimuli, and the white dashed circle indicates the initial position of the nanobubble. Scale bar: 50 nm.

Added discussions 2: Merging of neighboring of wrinkled nanopockets. We also agree with the reviewer that the earlier version of our work has not sufficiently discussed how the wrinkling affects the merging of nanopockets. To address so, in the revised version, we have included both qualitative AFM images of merging nanopockets and quantitative results on the degree of asymmetry of nanopockets with and without pretension. For example, in Fig. R12 above (also the new Fig. 4 in the manuscript), we show that when wrinkling connections form between two nanopockets, the nanopocket-A would move towards the nanopocket-B though the probe scanning direction shown by the white arrow is from bottom to top. This result indicates one of the key messages of our work that the nanopocket-nanopocket interaction exists between the neighboring nanopockets, which is particularly significant for those that are connected by wrinkling patterns. These discussions have been added in the revised version of our work (specifically, Lines: 219-233 in Manuscript; Lines: 130-135 in SI).

Fig. R13 (a-c) Movement of nanopockets subject to pretension. The white arrows correspond to the directions of mechanical stimuli, and the white dashed circle corresponds to the initial position of the nanobubble. Scale bars: 50 nm.

Added discussions 3: Merging of neighboring of unwrinkled nanopockets. When the pretension on nanopockets increases, it is found that the wrinkled region appearing at the pretension-free state starts to shrink and finally disappears. We further investigate the merging of pretensioned nanopockets whose wrinkles have been suppressed (Fig. R13 above, also Fig. 4 in the main text). We show in Fig. R13 that the nanopocket-F moves along the direction of probe scanning rather than toward the nanopocket-E. This observation actually agrees with one of our previous theoretical conclusions that the pretension suppresses the formation of wrinkling so that the pocket-pocket interaction is somewhat degraded. Since the asymmetry of the nanopocket when interacting with another provides the driving force for the merging, we further provide the inner and outer angles of the asymmetric nanopockets in Fig. 4c to reveal how the pretension affects the degree of the asymmetry. These results have been discussed in the revised manuscript (Lines: 235-258).

The reviewer had the following specific concerns:

1) Authors transferred 2 layers of graphene over hole that lead to nanopockets (not new, observed previously many times, see Ref-18 and Ref-A)

Our response: The fabrication method is not new. However, the focus of this work is not the fabrication method. It is embarrassing that our own work [Ref-A, published in Aug. 2020] was not cited. We have cited it in the revised version of manuscript, Ref. 24, Line 62.

2) Authors claim this is the first report of AFM observation of Moiré patterns in suspended bilayer graphene, line# 77-79 (false claim, previously published by authors themselves and not cited – Ref-A)

Our response: We have revised this claim in the revised manuscript since Ref-A has been published.

3) Authors observed coalescence of nanopockets with contact AFM, they also claim it could be driven by AFM probe – not new, but authors spend lots of time discussing it, lines#89-110

Our response: We thank the reviewer for this criticism. Accordingly, we have modified the presentation for this observation with a particular focus on the comparison between the coalescence of wrinkled and stretched/unwrinkled nanopockets (which we think is new) in our revised manuscript (Lines: 219-233).

4) Authors elaborate a model in details to describe the shape of the bubble. Not new, not properly cited, 15/18

misrepresented – this model is the from paper of one of the authors, Ref 8 – paper cited in general context at the begging, but not as a source of the model. lines#120-153

Our response: Thanks for the comment. The configuration here (drop being confined between two sheets) is different from previous models built for drop confined between a substrate and a sheet. So, we did not really use previous models, but we agree with the reviewer to recall previous bubble models properly. Besides, a particular difference between the present model and previous bubble model comes from the highly deformable boundary of the suspended nanopocket. The deformed boundary not only allows interesting wrinkling patterns surrounding bubbles but also enables fairly effective bubble-bubble interactions, which could further make self-cleaning possible. Theoretically, though the picture looks simple, it is a nontrivial task to understand the complex coupling of elastic instabilities, surface forces, and elastic forces in our system. Though the modeling has constituted three sections in the SI, we feel there remain a number of important subtleties to be investigated, such as wrinkling numbers, the force balance at the contact line when wrinkles exist (we for simplicity used the ad hoc force balance, which we believe to be a good approximation but needs a more rigorous proof) and so on. Due to the important complexity caused by the new physical ingredients such as instabilities and pretension effects, we think that the theoretical setting here is not just an incremental extension of previous ones.

5) Authors theoretically correlated graphene-graphene interface energy as a function of twist angle (novel, but it is incremental improvement on the above model).

Our response: We did not provide a theoretical prediction about interface energy as a function of the twist angle. The angle-dependent aspect ratios of nanopockets, instead of being a result of our theory, is part of experimental results.

6) Experimentally it is impossible to see if the above point holds. (Figure 2 is hard to read, but does not explicitly reveal any trend in h/a vs. twist angle) Even if it shows some trend, it would not be surprising, since the effect of twist angle on interfacial energy has been observed before (Ref 31, 32) lines#150-158

Our response: We thank the reviewer for this comment. To clarify, we show a separate figure (Figure 2b, line 136 in manuscript) to illustrate the average aspect ratios as a function of the twist angle (Fig. R14). Though the aspect ratio is a signature of the change of interfacial energies (or graphene-graphene adhesion), we did not translate it to the detailed graphene-graphene adhesion. The concerns are as follows: 1) The prefactor for such translation is not explicit due to the wrinkling. 2) To graphene-graphene adhesion, we need some information about the graphene-liquid interactions (such as contact angle or energy density), which would also be subject to uncertainties when wrinkles exist. 3) Knowing the detailed graphene-graphene adhesion should be important in many areas but a bit out of the focus of the current work. (Lines: 136-140 in Manuscript)

Fig. R14 Twist angle dependency of aspect ratios. Here, the data were collected from Fig. 2a in manuscript.

7) Authors apply external pressure on graphene membrane (not new), and observe effect of external tension on the shape of nanopockets (new—both experimentally and theoretically, but not super-exciting) lines #175-192.

Our response: Thanks for this comment. It is relatively intuitive that the nanopockets appear flatter after more pretension is applied. However, we think it is interesting to see the suppression of wrinkling by such pretension. (Lines: 180-198, Manuscript) and (Lines: 120-127, SI)

8) Authors explore experimentally wrinkling around nanopockets (new and interesting, but anecdotal) They show several images, and of wrinkling (and lack of), but they do not quantify the effect. It is not clear if this effect is artefact of AFM imaging or not.

Our response: Thanks for this criticism. In the revised manuscript, we have made more quantitative discussions regarding the formation of wrinkling for which the experimental results are collected and compared with the criterion predicted by our theory. These can be found in Fig. 3a in the manuscript and our response to your earlier comments: “*Added discussions 1: Suppression of wrinkling patterns*”. In addition, we think it is sure that the observation of these wrinkling patterns is quite robust rather than from any artifact.

9) They postulate the wrinkling is due to strain field around the bubble, which leads to long-range nanopocket interaction and their coalescence. The claim is not supported by data. No clear trend of wrinkling vs. coalescence is established (Fig3b,c have opposite trends than expected)

Our response: Thanks for this criticism. In the revised manuscript, we have provided a number of new experimental results and dedicated a new Figure 4 to clarify the pocket-pocket interactions (Lines: 235-258). These are also mentioned in our response to your earlier comments: “*Added discussions 2: Merging of neighboring of wrinkled nanopockets*” and “*Added discussions 3: Merging of neighboring of unwrinkled nanopockets*”.

10) They theoretically correlated appearance of instabilities (wrinkling) with geometry of nanopockets and external strain (novel), but there is no proper quantitative comparison with experiments (because experiments are not properly quantified) Some observations are off predictions, but they are waved away by hidden variables. Lines# 197-202

Our response: Thanks for this comment. In Figure 3a in our revised manuscript, we have used full markers and crossed markers to differentiate wrinkled and unwrinkled samples. We further prove that this

phenomenon is independent of the contaminants and the twist angle of tBLG in Figure 3b-e in manuscript (Lines: 160-170, 197-198). This could be used to compare the criterion for the two regimes as mentioned in our response to your earlier comments: “*Added discussions 1: Suppression of wrinkling patterns*”.

11) There is no discussion how MD simulations match theoretical considerations lines#238-256. The results (Fig4c) seem in contradictions with the theoretically predicted scaling (lines 219-224)

Our response: We thank the reviewer for pointing out this. Though it is challenging to quantitatively match between MD simulations and theories, we did find qualitative agreements between them in a few points, including those we have emphasized many times, such as the pretension effect on the nanopocket geometry and the loss of symmetry when nanopockets are interacting with one another. We are sorry that our previous presentation might be unclear and have made the reviewer feel “seem in contradictions”. In the revised manuscript, we have modified the discussion section to clarify how the simulation results could consolidate our claims. Specifically, we discussed the pretension effects on the nanopocket geometry in Lines: 296-300 and the symmetry loss in Lines: 284-295 (Manuscript).

12) The claims in the title and abstract should be focused in graphene, not general “van der Waals materials”. Some of the effect, including wrinkling and lubrication, might not generalize to the full extent in the other 2D materials (improper generalization)

Our response: We have changed the title to be “Elastocapillary cleaning of twisted bilayer graphene interfaces”.

13) One concerning issue is that the paper has 4 corresponding authors and 3 equal-contribution authors, while the volume of work is not much different that in most of the papers. This represents a strong inflation of the authorship contributions, and should have a high bar of justification.

Our response: This work includes combined efforts from a number of small-scale experiments, theories, and simulations. We have stated the authors' contributions at the end of the manuscript (Line: 372).

Extra references

A) Hou, Y. et al. Preparation of Twisted Bilayer Graphene via the Wetting Transfer Method. ACS Appl. Mater. Interfaces 12, 40958–40967 (2020)

Our response: We thank the reviewer for pointing out this reference and we have cited it in the manuscript (Ref. 24, Line 62).

Reviewers' Comments:

Reviewer #1:

Remarks to the Author:

I appreciate the efforts that the authors have dedicated to address my comments. I am still intrigued by the coalescence the authors observe and the proposed mechanical mechanism. It seems to me that such a study would motivate further work in a growing community interested in thin sheets mechanics. Nevertheless the remarks on novelty raised by the other referees are also important for a publication in Nature Communications.

Independently from the final editorial decision, reading again the revised manuscript, I got confused by the comparison of figure 4 with MD simulations (Fig 5d). From figure 4, I understand that the attraction between droplets decreases when pretension is applied. However MD simulations show that the drop in potential energy is larger for higher tensions. Is not it contradictory?

Another point: The authors mention the absence of water channels in the wrinkled configuration. However, would it be possible that the wrinkles behave as channels? Might it explain why coalescence is so quick in wrinkled configurations.

Reviewer #2:

Remarks to the Author:

.

Reviewer #3:

Remarks to the Author:

The authors spend lots of additional efforts to address the concerns raised by the first version of the manuscript – to their credit. There are few small questions that remain. If addressed, I believe this paper could be considered for publication.

== Figure 2a: ==

There is no reason why the figure should be presented as a bar chart. It would be more informative to present it as a regular graph.

== Figure 4. and line 248-258 ==

The authors use the contact angles of 2 interacting nanobubbles to prove the interaction between the bubbles.

Looking at the Fig 4a, the contact angle will depend on direction (angle) of the dashed line compared to the pocket. The authors should indicate that the line is taken through the centers of the two pockets.

Alternative, and way more instructive, approach could be to define the “roundness” of the nanobubble ([https://en.wikipedia.org/wiki/Roundness_\(object\)](https://en.wikipedia.org/wiki/Roundness_(object))) and demonstrate that the deviation from the round object leads to long-range interaction and coalescence of the objects. This graph could be added in either as the alternative to the Fig 4c or Fig 4d, or in supplementary information.

Moreover, the authors should try to correlate the non-roundness of neighboring nanopockets to the distance that separates them. The scaling of the curve ($1/\text{Roundness}$ or $1 - \text{Roundness}$) as a function of the separation distance, using all 19 data points of Fig4c may show the pocket interactions and distortions more efficiently than the molecular dynamics section. Some rescaling by the geometry may be necessary but this analysis should make the paragraph concerning the symmetry breaking more convincing (L248-258).

== Fig 5 ==

The energy scale seems very small: 1/10000 eV/atom. This is considerably smaller than kT (1/40 eV), so please comment on the energy scale and explain how pockets do not merge. Also, why are the potential energies corresponding to the snapshots of Fig 5b not shown in Fig 5d: what happens for $2a < d < 3a$? And below ($d < 2a$), the definition needs to be clarified.

== Fig S11 ==

"We found that the aspect ratios of ethanol nanopockets are ~ 0.106 , which is ~ 0.108 for water nanopockets." Please give +/- uncertainty values, the difference between ethanol and water is considerably smaller than the standard deviation.

== line 358 ==

The typo "pressure of ~ 0 MPa" is still there, despite reviewer 1's comment.

Reviewer #4:

Remarks to the Author:

In this work, Hou et al. investigated the coalescence of nanopockets confined between two graphene sheets, which is referred to as elastocapillary cleaning. This phenomenon was then interpreted as the role of stretch stress caused by the local deformation. I believe this is still a hot topic and definitely worth of further study. The present research made important progress in this field. This manuscript is well prepared. I can recommend its publication on Nature Communications. Here are specific comments.

1. After reviewing this revised manuscript, I found Reviewer 1 in the first round has raised some questions that I was intend to ask. And I think the authors have provided satisfied answers.
2. The experimental design is delicate, which allows for the measurement of the three-dimensional geometry of nanopockets AFM.
3. It is a bit confused in Fig. 5c, which was stated to be potential energy landscape. However, I noticed that the color bar represents a range from -2 Å to 2 Å. How could such height (deformation?) relate to the potential energy? It is not clear for me.
4. Continue with the last question. Is it possible to visualize the tension stress distribution of graphene layers in MD simulations? especially around the pockets. If yes, that would be great.
5. The authors wrote "The distance between the droplets d (along the zigzag direction in the graphene bilayer) is controlled through their centers of mass, which are tethered to specified positions through an elastic spring with a spring constant of 1600 N/m." Why d is controlled? and how? It seems that the coalescence of nanopockets is not spontaneous in MD simulations.

REVIEWER COMMENTS

Reviewer #1 (Remarks to the Author):

I appreciate the efforts that the authors have dedicated to address my comments. I am still intrigued by the coalescence the authors observe and the proposed mechanical mechanism. It seems to me that such a study would motivate further work in a growing community interested in thin sheets mechanics. Nevertheless, the remarks on novelty raised by the other referees are also important for a publication in Nature Communications.

Reply: We are grateful to the referee for the positive comments.

Independently from the final editorial decision, reading again the revised manuscript, I got confused by the comparison of Figure 4 with MD simulations (Fig 5d). From figure 4, I understand that the attraction between droplets decreases when pretension is applied. However MD simulations show that the drop in potential energy is larger for higher tensions. Is not it contradictory?

Reply: Thanks for this follow-up question. We want to clarify that the coalescence process includes a first attraction stage ($d > 2.5a$, a is the radius of droplets) driven by the elastic restoration force, and a second spontaneous coalescence step ($d < 2.5a$) after the initial contact is achieved between the two droplets. The experimental measurement captures the attraction stage. From **Fig. R1b** (see below), relative changes in the potential energies for the whole process are summarized by using the energy with merged droplets as references. The potential energy gradient in the attraction stage is reduced under higher tension, in consistency with experimental evidences. The spontaneous merging step, however, is driven by the cohesion of water droplet and cannot be resolved by the measurement. These results and discussions were added to **Fig. 5** in manuscript. (Lines 299-302, 313-316, 318-320 in Manuscript)

Fig. R1. (a) Molecular simulation snapshots showing the coalescence process of nanodroplets intercalated in a graphene bilayer with or without pretension. The atoms are colored by their

position out of the bilayer plane, where the color range is set to be smaller than the height of blisters to show the fringe of nanopockets and wrinkle patterns. (b) Potential energy landscape of the nanopockets as the inter-pocket distance d changes, where $a = 4$ nm is the radius of nanodroplet, and spring constant of $k = 1600$ N/m. The reference potential energy (at $d = 0$) is -4.678×10^6 , -4.674×10^6 and -4.661×10^6 eV for 0, 1% and 2% pre-stretch, respectively.

Another point: The authors mention the absence of water channels in the wrinkled configuration. However, would it be possible that the wrinkles behave as channels? Might it explain why coalescence is so quick in wrinkled configurations.

Reply: The wrinkles in the simulations are shown in **Fig. R2a**, suggesting that the wrinkles do not behave as channels before the spontaneous merging process. Based on the experimental characterization, **Fig. R2b-d** shows the lateral force images of nanopockets, where the presence of distorted moiré patterns in the wrinkled region suggest that there is no water molecules in the wrinkled region. The height profiles in **Fig. R2e-f** further reveal the morphologies of wrinkles between nanopockets. The result was added to **Supplementary Fig. 14**. (Lines 124-126 in SI)

Fig. R2. (a) Simulation side views of the wrinkles and droplets, which show that the wrinkles do not behave as channels before the droplets merge spontaneously. (b-d) The lateral force images of graphene nanopockets. (e-f) The cross-sectional height profiles of wrinkled regions. Scale bars: (b) 200 nm, (c, d) 50 nm.

Reviewer #3 (Remarks to the Author):

The authors spend lots of additional efforts to address the concerns raised by the first version of the manuscript – to their credit. There are few small questions that remain. If addressed, I believe this paper could be considered for publication.

Reply: We are grateful to the Referee for the enthusiastic comments and useful suggestions.

== Figure 2b: ==

There is no reason why the figure should be presented as a bar chart. It would be more informative to present it as a regular graph.

Reply: As suggested, **Fig. 2b** is revised to be a regular graph. (Line 136 in Manuscript)

== Figure 4. and line 248-258 ==

The authors use the contact angles of 2 interacting nanobubbles to prove the interaction between the bubbles.

Looking at the Fig 4a, the contact angle will depend on direction (angle) of the dashed line compared to the pocket. The authors should indicate that the line is taken through the centers of the two pockets.

Reply: We added the description about how these lines are made in the caption of **Fig. 4** in the manuscript. (Line 225 in Manuscript)

Alternative, and way more instructive, approach could be to define the “roundness” of the nanobubble ([https://en.wikipedia.org/wiki/Roundness_\(object\)](https://en.wikipedia.org/wiki/Roundness_(object))) and demonstrate that the deviation from the round object leads to long-range interaction and coalescence of the objects. This graph could be added in either as the alternative to the Fig 4c or Fig 4d, or in supplementary information.

Reply: We thank the Reviewer for this useful comment. According to the definition of the “roundness”, we define the “roundness” as the ratio of inscribed circle (r) and circumscribed circle (R) of the nanopocket, as shown in **Fig. R3a-b**. From the lateral force images of nanopockets, we found that the wrinkles around the two nanopockets are overlapped to form a connecting structure (we define them as “adjacent nanopockets”). The nanopockets far away from each other are defined as “single nanopockets”. The normalized roundness of the adjacent and single nanopockets with different sizes and pretensions are calculated. Here, the normalized roundness is expressed as r/R . **Fig. R3c-d** indicates that i) the normalized roundness of nanopockets is insensitive to the radii of nanopockets; ii) the normalized roundness of adjacent nanopockets is smaller than that of single nanopockets; iii) the pretension can decrease the

roundness of nanopockets. The roundness represents the proximity of nanopocket's shape to a standard circle, we infer that the distance between nanopockets and the pretension can influence the shape of nanopockets, which can be interpreted by the elastocapillarity cleaning mechanism. These results further support the conclusions of **Fig. 4** in the manuscript. We have added the results of roundness in Supplementary Information (**Supplementary Fig. 15**) and related discussion in the main text. (Lines 128-144 in SI; Lines 260-262 in Manuscript)

Fig. R3. (a-b) The lateral force images and schematics of adjacent and single nanopockets. (c-d) Roundness of adjacent/single nanopockets with and without pretension. Scale bars: (a)(b) 30 nm.

Moreover, the authors should try to correlate the non-roundness of neighboring nanopockets to the distance that separates them. The scaling of the curve ($1/\text{Roundness}$ or $1 - \text{Roundness}$) as a function of the separation distance, using all 19 data points of Fig4c may show the pocket interactions and distortions more efficiently than the molecular dynamics section. Some rescaling by the geometry may be necessary but this analysis should make the paragraph concerning the symmetry breaking more convincing (L248-258).

Reply: We appreciate the Reviewer's suggestion. To correlate the non-roundness of neighboring nanopockets to the distance, we define the normalized distance as $d/(R_L + R_S)$ and plot **Fig. R4**. To clarify the result, we consistently focus on the small nanopockets (R_S, r_S). It does suggest clearly that as the distance between nanopockets decreases, the roundness of the

smaller nanopockets shows a decrease trend. In the revised manuscript, we have added **Fig. R4** as part of **Supplementary Fig. 15**.

Fig. R4. The relationship between the roundness of nanopockets and the distance between nanopockets.

== Fig 5 ==

The energy scale seems very small: 1/10000 eV/atom. This is considerably smaller than kT (1/40 eV), so please comment on the energy scale and explain how pockets do not merge. Also, why are the potential energies corresponding to the snapshots of Fig 5b not shown in Fig 5d: what happens for $2a < d < 3a$? And below ($d < 2a$), the definition needs to be clarified.

Reply: We should clarify that the characteristic energy scale should not be calculated in the per-atom way, for the elastic distortion due to the droplets and their coalescence processes are localized. Lattice away from the droplets and wrinkles are not deformed by the inclusion. In this revision, we use the total energy instead for better illustration, although the value depends on the size of the simulated system.

The coalescence process includes a first attraction stage ($d > 2.5a$, a is the radius of droplets) driven by the elastic restoration force, and a second spontaneous coalescence step ($d < 2.5a$) after the initial contact is achieved between the two droplets. In **Fig. 5d**, relative changes in the potential energies for the whole process are summarized by using the energy with merged droplets as references. The potential energy gradient in the attraction stage is reduced under higher tension, in consistency with experimental evidences. The spontaneous merging step, however, is driven by the cohesion of water droplet and cannot be resolved by the measurement. **Fig. 5b** includes snapshots of molecular dynamics simulations, while the potential energy profile in **Fig. 5d** is obtained from free energy analysis through constrained energy minimization (see the updated Methods section for details). (Lines 299-302, 313-316, 318-320 in Manuscript)

== Fig S11 ==

“We found that the aspect ratios of ethanol nanopockets are ~ 0.106 , which is ~ 0.108 for water nanopockets.” Please give +/- uncertainty values, the difference between ethanol and water is considerably smaller than the standard deviation.

Reply: We have added the uncertainty values to **Supplementary Fig. 11**. (Line 106 in SI)

== line 358 ==

The typo “pressure of ~ 0 MPa” is still there, despite reviewer 1’s comment.

Reply: Thanks for spotting this, which was corrected in this revision.

Reviewer #4 (Remarks to the Author):

In this work, Hou et al. investigated the coalescence of nanopockets confined between two graphene sheets, which is referred to as elastocapillary cleaning. This phenomenon was then interpreted as the role of stretch stress caused by the local deformation. I believe this is still a hot topic and definitely worth of further study. The present research made important progress in this field. This manuscript is well prepared. I can recommend its publication on Nature Communications. Here are specific comments.

1. After reviewing this revised manuscript, I found Reviewer 1 in the first round has raised some questions that I was intend to ask. And I think the authors have provided satisfied answers.

Reply: We thank the Reviewer for these positive comments.

2. The experimental design is delicate, which allows for the measurement of the three-dimensional geometry of nanopockets AFM.

Reply: We are grateful to the Reviewer for this comment.

3. It is a bit confused in Fig. 5c, which was stated to be potential energy landscape. However, I noticed that the color bar represents a range from -2 Å to 2 Å. How could such height (deformation?) relate to the potential energy? It is not clear for me.

Reply: **Fig. 5c** in the manuscript shows the out-of-plane displacement of carbon atoms in the top graphene layer to show the edge of droplets and wrinkle patterns, where the height is about 0.13-0.15 of the radius of the droplets. The definition is elaborated in **Fig. R1**. (Lines 272-273 in Manuscript)

4. Continue with the last question. Is it possible to visualize the tension stress distribution of graphene layers in MD simulations? especially around the pockets. If yes, that would be great.

Reply: Thank you for the suggestions. The average stress $(\sigma_1 + \sigma_2 + \sigma_3)/3$ is calculated, showing that the wrinkled features identified in the absence of pretension are reduced by applying pretension of 2%. (see **Fig. R5** and **Supplementary Fig. 17**, Lines 153-156 in SI).

Fig. R5. Stress distributions during coalescence of nanopockets with and without pretension.

5. The authors wrote “The distance between the droplets d (along the zigzag direction in the graphene bilayer) is controlled through their centers of mass, which are tethered to specified positions through an elastic spring with a spring constant of 1600 N/m.” Why d is controlled? and how? It seems that the coalescence of nanopockets is not spontaneous in MD simulations.

Reply: We want to clarify that the coalescence process includes a first attraction stage ($d > 2.5a$, a is the radius of droplets) driven by the elastic restoration force, and a second spontaneous coalescence step ($d < 2.5a$) after the initial contact is achieved between the two droplets. We controlled the value of d in our constrained energy minimization calculations to analyze the potential energy profile only (see the updated Methods section for details).

Snapshots in **Fig. 5b** are obtained from MD simulations where the spontaneous processes of coalescence are identified. However, the small size of simulated systems here results in relatively strong fluctuation in the kinetic and potential energy terms. To exclude the fluctuation and quantify the driving force, potential energy evolution during the coalescence is analyzed using constrained energy minimization, where the center of mass is constrained as kR_cM_i/M (k is the spring constant, R_c is the position of the center of mass to be fixed, M_i and M are the mass of single carbon atom and the whole system, respectively). The spring constant k is a key parameter that could modify the potential energy landscape and modify the merging process. For a large value of $k = 1600$ N/m, the fluctuation of R_c is not significant, and the spontaneous merging processes are observed as the inter-droplet distance decreases to $2.5a$, where $a = 4$ nm is the radius of droplets. For a small value of $k = 0.16$ N/m, the fluctuation is significant (± 2 nm). The measured potential energy profile is thus more noisy, although the main features remain the same (**Fig. R6**). These details were added to the **Methods** section (Lines 371-385 in Manuscript). **Movie S1** was added to show the spontaneous process of coalescence. (Lines 299-302, 313-316, 318-320 in Manuscript)

Fig. R6. Effects of spring stiffness k on the potential energy landscape of nanopocket-containing graphene bilayer, which is analyzed as a function of the inter-pocket distance d .

REVIEWERS' COMMENTS

Reviewer #1 (Remarks to the Author):

It seems to me that the authors addressed the different comment the referees raised. Appreciable improvements and clarifications have been incorporated into the manuscript.
I would recommend the publication of the current version to Nature Communications.

I found a few typos in the manuscript:

Main text:

- line 220: the wrinkled region shrinks and finally disappears, which are shown in Fig. 3d and e. -> I would write "which is shown" (but I may be wrong)
- line 222: water and ethanol nanopocket -> nanopockets
- line 233: after the second mechanical stimuli -> stimulus ?
- line 378: investigated by stretch -> by stretching
- line 379: To exclude the fluctuation -> fluctuations
- line 388: s thus more noisy -> noisier

Supplementary information:

I found difficult to read the axis titles in Fig. 15.

- line 317: We introduced two unknown -> unknowns

Reviewer #4 (Remarks to the Author):

I have previously reviewed the manuscript and the authors have addressed all my concerns. I recommend the publication of the manuscript in its current state.

REVIEWER COMMENTS

Reviewer #1 (Remarks to the Author):

It seems to me that the authors addressed the different comment the referees raised. Appreciable improvements and clarifications have been incorporated into the manuscript.

I would recommend the publication of the current version to Nature Communications.

Reply: We are grateful to the referee for the positive comments.

I found a few typos in the manuscript:

Main text:

- line 220: the wrinkled region shrinks and finally disappears, which are shown in Fig. 3d and e. -> I would write "which is shown" (but I may be wrong)
- line 222: water and ethanol nanopocket -> nanopockets
- line 233: after the second mechanical stimuli -> stimulus ?
- line 378: investigated by stretch -> by stretching
- line 379: To exclude the fluctuation -> fluctuations
- line 388: s thus more noisy -> noisier

Reply: Thanks for the comments. These typos have been revised. (Lines: 223, 225, 235, 389, 390, 399 in Manuscript)

Supplementary information:

I found difficult to read the axis titles in Fig. 15.

- line 317: We introduced two unknown -> unknowns

Reply: Thanks for the comments. The axis titles in Fig. 15 have been magnified. This typo has been revised. (Line: 317 in SI)

Reviewer #4 (Remarks to the Author):

I have previously reviewed the manuscript and the authors have addressed all my concerns. I recommend the publication of the manuscript in its current state.

Reply: We are grateful to the referee for the positive comments.